# Unifying Structure- and Ligand-based Drug Design via Contrastive Geometric Learning

## Abstract

Structure-based computational drug design, which employs methods trained on large datasets of protein–ligand complex structures, has been revolutionized by breakthroughs such as AlphaFold. In parallel, ligand-based computational drug design, driven by models trained on extensive bioactivity resources, has impacted drug discovery by enabling the simultaneous prediction of numerous biological effects of small-molecule ligands. Yet, despite recent advances in both structure- and ligand-based approaches, no existing method integrates them effectively at scale. We introduce **Con**trastive **G**eometric **L**earning for **U**nified Computational **D**rug D**e**sign (ConGLUDe), an approach that leverages both structure- and ligand-based training data through geometric and contrastive learning. The ConGLUDe architecture combines a geometric protein encoder, producing both spatial binding pocket and global protein representations, with a ligand encoder. The encoders are trained jointly via contrastive learning on 20K protein–ligand complexes from PDB-bind and 77M ligand-based datapoints from ChEMBL, PubChem, and BindingDB. With ConGLUDe, multiple key drug discovery tasks, including virtual screening, binding pocket prediction, ligand-conditioned pocket selection and target fishing, can be addressed within a single model. ConGLUDe achieves state-of-the-art performance on zero-shot virtual screening benchmarks and strong results across other tasks, demonstrating the benefit of joint structure–ligand training. By replacing a set of specialized models with a single system and by unifying structure- and ligand-based paradigms, ConGLUDe represents a major step toward foundation models for drug discovery.

## 1 Introduction

**The key component of drug discovery is the interaction between a protein and a potential ligand.** Most drugs are small molecules that bind to a disease-associated protein target to activate, inhibit, or modify its function (Kinch et al., 2024). Understanding these protein-ligand interactions (PLIs) enables meaningful engagement with biological systems and the purposeful design of therapeutic agents (Gohlke et al., 2000; Du et al., 2016). For decades, computational methods, collectively referred to as computer-aided drug design (CADD), have been employed to predict and analyze these interactions. These computational methods have traditionally been categorized into two primary paradigms: structure-based drug design (SBDD) and ligand-based drug design (LBDD), depending on whether the methods approach the PLI problem via the protein structure or ligand activities (Macalino et al., 2015; Vemula et al., 2023). In recent years, advancements in artificial intelligence (AI) and machine learning (ML) have profoundly enhanced the understanding and modeling of protein-ligand interactions. These technologies have been applied directly in both LBDD (Dahl et al., 2014; Lenselink et al., 2017; Mayr et al., 2018) and SBDD (Ballester & Mitchell, 2010b; Corso et al., 2023) methods, as well as indirectly through breakthroughs in protein modeling. Notably, developments like AlphaFold have revolutionized protein structure prediction, "enabling" SBDD for any protein sequence and significantly advancing the design of novel therapeutics (Jumper et al., 2021).

**Structure-based and ligand-based drug design are the two fundamental paradigms of drug discovery.** *Structure-based drug design (SBDD)* (Blundell, 1996) relies on the three-dimensional (3D) structure of the target protein's binding site. Generally, this information is obtained through the experimental determination of protein-ligand complexes (Mutharasappan et al., 2020), a process

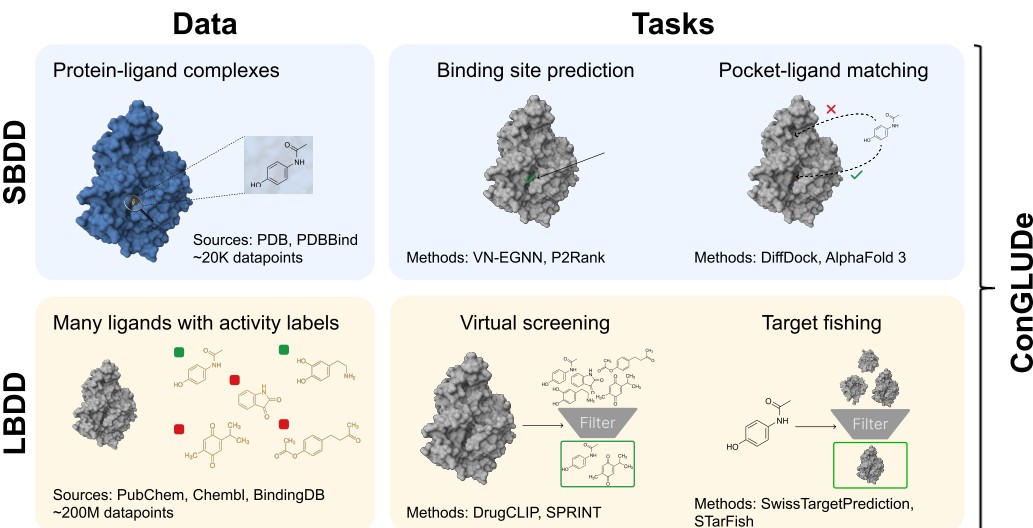

Figure 1: ConGLUDe unifies structure-based and ligand-based drug design.

that is far from trivial and has historically limited the application of SBDD to only a fraction of known proteins. Experimental structures are systematically archived in the Protein Data Bank (PDB) (Berman et al., 2000), which contains ~235k entries[1], a fraction of which includes biologically relevant ligands, providing a valuable resource for SBDD research. Recently, the challenge of obtaining 3D protein structures has been largely addressed by AlphaFold2 (Jumper et al., 2021), which provides accurate predictions for most protein sequences. However, AlphaFold2 does not offer information about ligand binding sites, leaving binding site prediction as a crucial step in the SBDD pipeline (Zhao et al., 2020). Once the binding site is identified, candidate molecules are screened with methods such as docking (Kuntz et al., 1982; Fan et al., 2019), molecular dynamics (De Vivo et al., 2016), and free energy perturbation (Beveridge & Dicapua, 1989; Cournia et al., 2021) to evaluate their binding potential and understand protein-ligand interactions. Traditionally, molecular docking has relied on simple (semi-) empirical scoring functions to quantify these interactions (Li et al., 2019), but ML- and AI-based scoring functions have also emerged (Ballester & Mitchell, 2010b; Wallach et al., 2015). More recently, AI innovations have enabled the holistic prediction of protein-ligand complexes either with AI-based blind docking methods (Stärk et al., 2022; Corso et al., 2023; Pei et al., 2024), or foundation models for molecular structure prediction of biological complexes (Abramson et al., 2024; Wohlwend et al., 2024; Discovery et al., 2024).

*Ligand-based drug design (LBDD)* (Merz Jr et al., 2010) relies solely on ligand-based information and experimental data on ligand activity for a target of interest, without requiring knowledge of how the ligand interacts with the protein. A large amount of ligand-based data has been made publicly available in databases such as PubChem, which contains approximately 300 million bioactivity data points (Kim et al., 2024). This wealth of data has made ML an integral part of LBDD since the early 1990s, in the form of quantitative structure-activity relationship (QSAR) (Hansch et al., 1962; Muratov et al., 2020) modeling leveraging support vector machines (Burbidge et al., 2001), random forests (Svetnik et al., 2003), gradient boosting (Babajide Mustapha & Saeed, 2016; Sheridan et al., 2016), and more recently, multi-task deep neural networks (Lenselink et al., 2017; Mayr et al., 2018; Yang et al., 2019). Traditionally, ML-based LBDD has been limited to protein targets with sufficient experimental data to train target-specific QSAR models. However, recent few-shot and zero-shot learning methods have expanded activity prediction to scarce-data scenarios (Vella & Ebejer, 2022; Schimunek et al., 2023; Seidl et al., 2023). Proteochemometrics augments ligand-based models with explicit protein representations, typically sequence-derived descriptors such as physicochemical amino-acid scales or learned embeddings, so a single model can generalize across related targets, capture selectivity patterns and supports transfer to unseen targets (Lapinsh et al., 2001; Öztürk et al., 2018; Bongers et al., 2019; Svensson et al., 2024).

---

[1]From https://www.rcsb.org/stats/growth/growth-released-structures. Accessed on 08/05/2025.

**Structure-based approaches have weak ligand representations, and ligand-based approaches have weak protein-structure representations.** While both SBDD and LBDD have led to many successful drug discovery projects and continue to bridge biomolecular research with machine learning, neither paradigm fully exploits the complementary wealth of structural and ligand data needed to learn meaningful, joint representations (Sadybekov & Katritch, 2023). Within the structure-based methods, AlphaFold3 (Abramson et al., 2024) was trained on almost the entire PDB, which contains 200k protein structures, but contains only 40k small molecules (Shao et al., 2022), limiting the depth of the ligand representations. Conversely, ligand-based models such as ChemNet (Preuer et al., 2018), trained on 220 million bioactivity measurements covering 3.6 million compounds, or transformer architectures like ChemBERTa (Chithrananda et al., 2020) and MolBERT (Li & Jiang, 2021), pretrained on 77 million and 4 billion SMILES strings, respectively, include no explicit protein structural information and therefore yield only weak, implicit representations of protein targets. Jointly training robust protein and ligand representations on a shared, biologically meaningful task promises to dramatically enhance drug discovery by capturing the intricate interplay of protein–ligand interactions. Yet, no existing architecture simultaneously leverages both the full breadth of three-dimensional structural data (e.g., tens of thousands of PDB entries) and large-scale ligand databases (e.g., millions of bioactivity measurements) within a unified learning framework.

**Unification of structure- and ligand-based drug discovery data through geometric contrastive learning allows for foundation models in drug discovery.** We introduce *Contrastive Geometric Learning for Unified Computational Drug Design* (ConGLUDe), a framework that co-trains a geometric protein encoder, producing both spatial binding pocket and global protein representation, and a ligand encoder using contrastive objectives . This is the first framework that enables co-training a model on both structure-based data of 3D structures of protein–ligand complexes from the PDB and ligand-based bioactivity data from PubChem (Kim et al., 2024), BindingDB (Gilson et al., 2015), and ChEMBL (Gaulton et al., 2011), and allows a single model to be used for many different drug discovery tasks, such as a) virtual screening, b) binding pocket identification, c) ligand-conditioned pocket selection, and d) target fishing (Figure 1). In our evaluations, ConGLUDE attains state-of-the-art virtual screening, remains competitive for binding site detection, improves pocket selection by ligand-conditioning, and delivers impressive performance at zero-shot target fishing.

## 2 BACKGROUND AND PRELIMINARIES

### 2.1 NOTATION AND DEFINITIONS

**Protein–ligand interaction data point.** A PLI data point is defined as a triplet $(\mathcal{G}, \mathcal{M}, y)$, where $\mathcal{G}$ denotes a protein, $\mathcal{M}$ a ligand (typically a small molecule), and $y$ a binary or real-valued label. In structure-based datasets, PLIs data points are derived from experimentally resolved 3D structures of protein-ligand complexes. Protein–ligand pairs with observed co-crystal structures are labeled as positives ($y = 1$), while all other combinations are treated as negatives ($y = 0$). In contrast, ligand-based datasets provide activity measurements for a large set of small molecules tested against a given target protein, typically obtained through biological assays. Labels may be binary (active: $y = 1$, inactive: $y = 0$) or continuous affinity values ($y \in \mathbb{R}$), such as $IC_{50}$ or $K_d$.

**Protein and ligand representations.** We represent proteins as geometric graphs $\mathcal{G}$, where each node corresponds to an amino acid residue. Each node is assigned a 3D coordinate (specifically, the position of the $C_\alpha$ atom) and a feature vector encoding residue-specific properties, extracted using ESM-2 (Lin et al., 2023). Edges connect each node to a maximum of 10 nearest neighbors within a 10 Å radius. Ligands are represented as fixed-length vectors constructed by concatenating Morgan fingerprints (Morgan, 1965) with RDKit chemical descriptors (Landrum & contributors, 2006).

**Definition of binding sites.** Structure-based datasets enable direct annotation of protein binding sites – the regions where ligands interact with the protein. Here, we define a binding site for a given ligand as the geometric center $\mathbf{z} \in \mathbb{R}^3$ of all protein residues that lie within a 4 Å radius of any ligand atom.

An overview of all notation used in this work is provided in Appendix A.

## 2.2 Binding Pocket Prediction Using VN-EGNN

When experimental binding site annotations are unavailable, accurately identifying binding pockets becomes a critical step in SBDDs. Sestak et al. (2024) proposed an approach based on an equivariant graph neural network with virtual nodes (VN-EGNN) to address this task. In this framework, the protein is represented as a geometric graph (as described above), augmented with a small set of virtual nodes. Each virtual node is initialized with a coordinate on a sphere around the protein and a feature vector given by the mean of all protein residue embeddings. Virtual nodes are connected to every protein residue, enabling the network to integrate both local and global structural information. VN-EGNN employs a three-step heterogeneous message-passing scheme between protein and virtual nodes, detailed in Appendix C.1. The model is trained with a combination of three objective functions (see Appendix C.2) to predict the 3D coordinates of potential binding pockets, denoted by the final virtual node positions $\mathbf{z}'_1, \ldots, \mathbf{z}'_N \in \mathbb{R}^3$, where $N$ is the number of virtual nodes. In addition to predicting binding site centers, the model outputs pocket-level feature representations $\boldsymbol{b}'_1, \ldots, \boldsymbol{b}'_N \in \mathbb{R}^E$ from the final layer. These embeddings are used to assign confidence scores to predicted pockets and can facilitate downstream tasks such as pocket ranking or contrastive learning.

## 2.3 Virtual Screening Using Contrastive Learning

Contrastive learning has recently emerged as a powerful paradigm for virtual screening, enabling protein and ligand representations to be embedded in a shared latent space where interactions are inferred via representational similarity (Singh et al., 2023; Gao et al., 2024; Han et al., 2024; Wang et al., 2024; McNutt et al., 2024; Gil-Sorribes et al., 2025). This framework typically consists of three components:

- a *molecule encoder*, which projects ligand representations into the shared latent space,
- a *protein and pocket encoder*, which maps sequence- or structure-based representations of the target protein and binding site into the same space, and
- a *contrastive loss function*, which encourages interacting protein–ligand pairs to have similar embeddings and non-interacting pairs to be dissimilar.

Contrastive approaches have achieved state-of-the-art performance compared to traditional docking methods. A key advantage is their computational efficiency: embeddings can be precomputed, allowing large-scale screening to be reduced to fast similarity calculations between protein and ligand embeddings. Most existing methods rely either on whole-protein representations (Singh et al., 2023; Wang et al., 2024; McNutt et al., 2024) or on predefined binding pocket representations (Gao et al., 2024; Han et al., 2024). The recently introduced Tensor-DTI (Gil-Sorribes et al., 2025) combines both protein- and pocket-level encodings.

# 3 Contrastive-Geometric Learning for Unified Drug Design (ConGLUDe)

In short, ConGLUDe employs a geometric *protein encoder* based on a modified VN-EGNN (Sestak et al., 2024) architecture, which predicts candidate binding site locations $\hat{\mathbf{z}}_1, \ldots, \hat{\mathbf{z}}_K \in \mathbb{R}^3$ together with corresponding representations $\boldsymbol{b}_1, \ldots, \boldsymbol{b}_K \in \mathbb{R}^D$ as well as a global protein embedding $\boldsymbol{p} \in \mathbb{R}^D$. A complementary *molecule encoder* maps ligands into representations $\boldsymbol{m} \in \mathbb{R}^{2D}$, aligned with the concatenated protein/pocket embeddings $[\boldsymbol{b}_i, \boldsymbol{p}]$.

ConGLUDe integrates structure- and ligand-based learning by alternating between (i) structure-based batches, where it learns to detect and characterize binding sites and pair them with their ligands, and (ii) ligand-based batches, where it leverages large-scale bioactivity measurements. Figure 2 provides an overview of the architecture and training procedure.

## 3.1 ConGLUDe Architecture

### 3.1.1 Protein and Binding Pocket Encoders.

We extend the original VN-EGNN formulation by introducing an additional non-geometric virtual node $\mathcal{P}$, which aggregates information from the entire protein but has no spatial coordinates. In

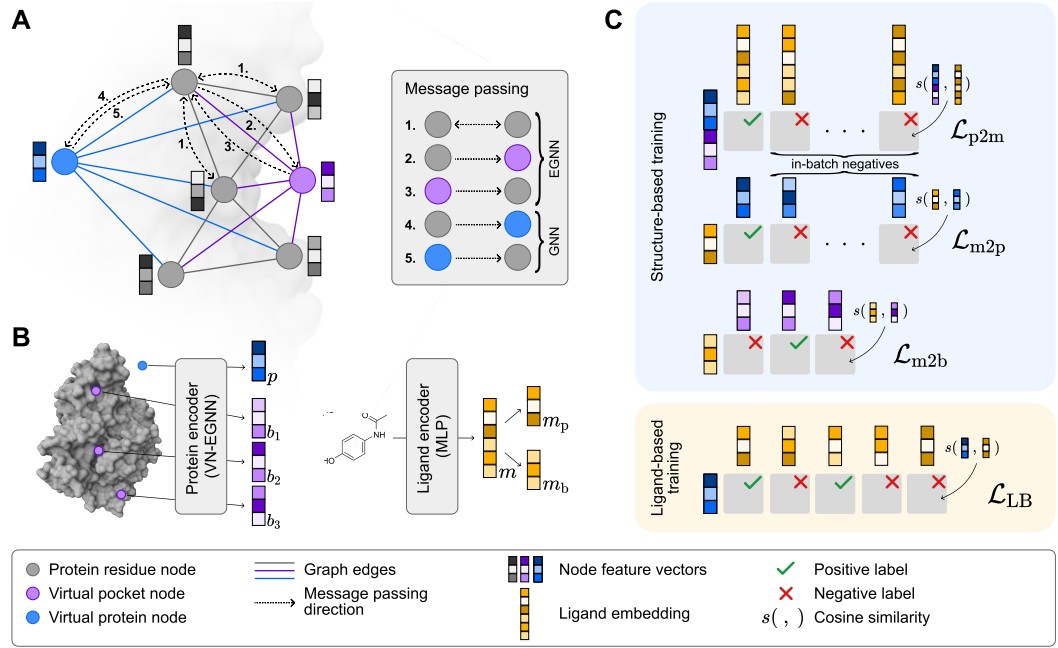

Figure 2: ConGLUDe architecture. **A**: Message-passing scheme of ConGLUDe with five steps: 1. message exchange between residue nodes, 2. residue nodes to virtual pocket nodes, 3. pocket nodes to residue nodes, 4. residue nodes to virtual protein node, 5. virtual protein node to residue nodes. **B**: The protein encoder supplies a representation of the whole protein $\boldsymbol{p}$, and of each detected pocket $\boldsymbol{b}_k$. The ligand encoder encodes each small molecule into a protein matching representation $\boldsymbol{m}_{\mathrm{p}}$ and a pocket-matching representation $\boldsymbol{m}_{\mathrm{b}}$. **C**: Contrastive loss functions used in our approach. Structure-based losses include $\mathcal{L}_{\mathrm{m2p}}$: InfoNCE between a concatenated protein-pocket representation and all ligand representations from the batch, $\mathcal{L}_{\mathrm{p2m}}$: InfoNCE between a ligand and all protein representations in the batch, and $\mathcal{L}_{\mathrm{m2b}}$: InfoNCE between a ligand and all pocket representations from the corresponding protein. The NCE loss between a protein and annotated ligand representations ($\mathcal{L}_{\mathrm{LB}}$) is used on ligand-based data.

addition to the three geometric message-passing steps of VN-EGNN (Appendix C.1), we add two non-geometric steps from residue nodes to the protein node ($\mathcal{R} \rightarrow \mathcal{P}$) and vice versa. ($\mathcal{P} \rightarrow \mathcal{R}$):

**Message passing step 4 ($\mathcal{R} \rightarrow \mathcal{P}$):**      **Message passing step 5 ($\mathcal{P} \rightarrow \mathcal{R}$):**

$$\boldsymbol{\mu}_j^{(\mathcal{RP})} = \boldsymbol{\phi}_{e(\mathcal{RP})}(\boldsymbol{p}, \boldsymbol{h}_j) \qquad (1)$$

$$\boldsymbol{\mu}^{(\mathcal{RP})} = \frac{1}{S} \sum_{j=1}^{S} \boldsymbol{\mu}_j^{(\mathcal{RP})} \qquad (2)$$

$$\boldsymbol{\mu}_i^{(\mathcal{PR})} = \boldsymbol{\phi}_{e(\mathcal{PR})}(\boldsymbol{h}_i, \boldsymbol{p}) \qquad (4)$$

$$\boldsymbol{h}_i = \boldsymbol{h}_i + \boldsymbol{\phi}_{h(\mathcal{BR})}\left(\boldsymbol{h}_i, \boldsymbol{\mu}_i^{(\mathcal{PR})}\right) \qquad (5)$$

$$\boldsymbol{p} = \boldsymbol{p} + \boldsymbol{\phi}_{h(\mathcal{RP})}\left(\boldsymbol{p}, \boldsymbol{\mu}^{(\mathcal{RP})}\right) \qquad (3)$$

Here, $\boldsymbol{\mu}_j^{(\mathcal{RP})}$ denotes the messages sent from residue node $j$ to the protein node, while $\boldsymbol{\mu}_i^{(\mathcal{PR})}$ denotes the reverse direction. The functions $\boldsymbol{\phi}_{e(\mathcal{RP})}, \boldsymbol{\phi}_{h(\mathcal{RP})}, \boldsymbol{\phi}_{e(\mathcal{PR})}$ and $\boldsymbol{\phi}_{h(\mathcal{BR})}$ are layer-specific multi-layer-perceptrons (MLPs) of the GNN. Our model uses 5 layers of VN-EGNN, but we omit the layer index in Eq. C.1–C.12 and Eq.1–5 for clarity. Applying the structure encoder to a protein graph $\mathcal{G}$ yields

$$\mathbf{X}', \boldsymbol{H}', \mathbf{Z}', \boldsymbol{B}', \boldsymbol{p}' = \mathrm{VNEGNN}(\mathcal{G}),$$

where $\mathbf{X}' = (\mathbf{x}_1', \dots, \mathbf{x}_S') \in \mathbb{R}^{S \times 3}$ and $\boldsymbol{H}' = (\boldsymbol{h}_1', \dots, \boldsymbol{h}_S') \in \mathbb{R}^{S \times E}$ are the residue coordinates and features, $\mathbf{Z}' = (\mathbf{z}_1', \dots, \mathbf{z}_N') \in \mathbb{R}^{N \times 3}$ are the coordinates of the virtual nodes representing binding pockets, and $\boldsymbol{p}'$ is the global protein embedding from the protein virtual node.

To rank binding pocket predictions by confidence, we follow Sestak et al. (2024) and apply a two-layer MLP with scalar outputs to the pocket representations: $\boldsymbol{c}' = \mathrm{MLP}(\boldsymbol{B}'), \boldsymbol{c}' \in \mathbb{R}^N$. Since multiple virtual nodes may converge to the same binding pocket, we cluster them based on their spatial coordinates using DBSCAN (Ester et al., 1996). For each cluster, we then compute the mean of the coordinates, feature vectors, and confidence values, yielding $\hat{\mathbf{X}} \in \mathbb{R}^{K \times 3}, \hat{\boldsymbol{H}} \in \mathbb{R}^{K \times E}, \hat{\boldsymbol{c}} \in \mathbb{R}^K$ with $K < N$. Finally, pocket- and protein-level representations are projected into the contrastive embedding space of dimension $D$ via linear transformations: $\boldsymbol{B} = \mathrm{Linear}(\hat{\boldsymbol{B}}), \quad \boldsymbol{p} = \mathrm{Linear}(\boldsymbol{p}')$.

### 3.1.2 LIGAND ENCODER

For the ligand encoder, we adopt a simple yet effective design motivated by prior work, which has shown that molecular fingerprints combined with MLPs often outperform more complex architectures such as graph neural networks for encoding small molecules (Siemers et al., 2022; Luukkonen et al., 2023; Praski et al., 2025; Seidl et al., 2023; Unterthiner et al., 2014). Formally, the initial ligand representation is mapped into the contrastive embedding space of dimension $2D$ using a 2-layer MLP:

$$\boldsymbol{m} = [\boldsymbol{m}_{\mathrm{p}}, \boldsymbol{m}_{\mathrm{b}}] = \mathrm{MLP}(\mathcal{M}), \quad \boldsymbol{m} \in \mathbb{R}^{2D}.$$

This lightweight architecture enables simultaneous encoding of large batches of ligands, making it well-suited for high-throughput virtual screening across extensive compound libraries.

### 3.1.3 INFERENCE MODES

ConGLUDe supports multiple inference modes. In classical *virtual screening*, predictions are made by comparing the protein representation with the protein-specific component of the ligand embedding, $s(\boldsymbol{p}, \boldsymbol{m}_{\mathrm{p}})$, where $s(.,.)$ denotes the cosine similarity and higher similarity indicates a higher likelihood of binding. This formulation also applies to target fishing, where a ligand is tested across multiple proteins. For *binding site identification*, the VN-EGNN–based encoder directly outputs candidate pocket centers with confidence values. Predicted pockets can be ranked either ligand-independently by these confidence scores or in a ligand-conditioned manner by their similarity to the pocket-specific component of the ligand embedding, $s(\boldsymbol{b}_l, \boldsymbol{m}_{\mathrm{b}})$. When the objective is to evaluate ligand binding to a predefined pocket on a given protein, ConGLUDe employs a similarity measure between the ligand embedding and the protein–pocket representation, $s([\boldsymbol{p}, \boldsymbol{b}_l], \boldsymbol{m})$.

## 3.2 CONGLUDE TRAINING

### 3.2.1 DATA

The ConGLUDe model can be trained on a combination of both structure-based and ligand-based data. For each task, the structure-based training data, a subset of PDBBind by Wang et al. (2005), are derived from the respective baseline methods. As ligand-based data we use the MERGED dataset curated by McNutt et al. (2024), which combines PubChem (Kim et al., 2024), BindingDB (Gilson et al., 2015), and ChEMBL (Gaulton et al., 2011) and remove all proteins with >90% sequence identity to any test set protein. For details on all datasets, see Appendix C.

### 3.2.2 TRAINING OBJECTIVE

The ConGLUDe objective is to minimize the loss on both ligand-based and structure-based data:

$$\mathcal{L} = \mathcal{L}_{\mathrm{SB}} + \mathcal{L}_{\mathrm{LB}}, \tag{6}$$

where $\mathcal{L}_{\mathrm{SB}}$ is the loss on structure-based training data and $\mathcal{L}_{\mathrm{LB}}$ is the loss on ligand-based training data, which are detailed further below. During training, each step samples a batch of either structure-based or ligand-based data at random, and the optimization objective is applied accordingly.

**Training on Structure-Based Data.** For structure-based data, annotated protein binding sites provide supervision for binding site prediction. In this setting, the loss decomposes into a geometric term and a contrastive term:

$$\mathcal{L}_{\mathrm{SB}} = \mathcal{L}_{\mathrm{geometric}} + \mathcal{L}_{\mathrm{contrastive}}. \tag{7}$$

The geometric component, $\mathcal{L}_{\text{geometric}}$. is equivalent to the objective function of VN-EGNN (see Sestak et al. (2024) and Appendix C.2).

Beyond the geometric objective, we leverage contrastive learning to align the representations of ligands with their corresponding proteins and predicted binding pockets. For a given protein-ligand complex, the ligand embedding $\boldsymbol{m}^{(j)}$ is encouraged to be close in representation space to the concatenated protein and pocket embeddings $[\boldsymbol{p}^{(j)}, \boldsymbol{b}_l^{(j)}]$, where $\boldsymbol{b}_l^{(j)}$ is the predicted pocket closest to the ligand's true binding site: $l = \arg\min_{k=1,\ldots,K}(\|\mathbf{z} - \hat{\mathbf{z}}_k\|)$.

This alignment is implemented using a three-way InfoNCE loss, similar to CLIP (Radford et al., 2021):

$$\mathcal{L}_{\text{contrastive}} = \frac{1}{3J} \sum_{j=1}^{J} \left( \mathcal{L}_{\text{p2m}}^{(j)} + \mathcal{L}_{\text{m2p}}^{(j)} + \mathcal{L}_{\text{m2b}}^{(j)} \right), \text{with} \tag{8}$$

$$\mathcal{L}_{\text{p2m}}^{(j)} = \text{InfoNCE}([\boldsymbol{p}^{(j)}, \boldsymbol{b}_l^{(j)}], \boldsymbol{m}^{(j)}, \{\boldsymbol{m}^{(1)}, \ldots, \boldsymbol{m}^{(J)}\}; \tau_{\text{p2m}}), \tag{9}$$

$$\mathcal{L}_{\text{m2p}}^{(j)} = \text{InfoNCE}(\boldsymbol{m}_{\text{p}}^{(j)}, \boldsymbol{p}^{(j)}, \{\boldsymbol{p}^{(1)}, \ldots, \boldsymbol{p}^{(I)}\}; \tau_{\text{m2p}}), \text{and} \tag{10}$$

$$\mathcal{L}_{\text{m2b}}^{(j)} = \text{InfoNCE}(\boldsymbol{m}_{\text{b}}^{(j)}, \boldsymbol{b}_l^{(j)}, \{\boldsymbol{b}_1^{(j)}, \ldots, \boldsymbol{b}_K^{(j)}\}; \tau_{\text{m2b}}). \tag{11}$$

with the usual definition of InfoNCE (see Eq.A.1). In the first direction - "protein/pocket to molecule" – the protein/pocket representation acts as the anchor, and the model is trained to associate it with its true ligand while treating other ligands in the batch as negatives. In the reverse direction, the ligand representation is split into two components. The first component, $\boldsymbol{m}_{\text{p}}^{(j)}$, is aligned with the protein embedding $\boldsymbol{p}^{(j)}$ while contrasting it against other proteins in the mini-batch ("molecule to protein"). The second component, $\boldsymbol{m}_{\text{b}}^{(j)}$, is aligned with the closest predicted binding pocket $\boldsymbol{b}_l^{(j)}$ while contrasting it against the remaining pockets predicted on the same protein ("molecule to binding site"). The temperature parameters are chosen as the inverse square root of the corresponding contrastive space dimension, i.e. $\tau_{\text{p2m}} = \frac{1}{\sqrt{2D}}$ and $\tau_{\text{m2p}} = \tau_{\text{m2b}} = \frac{1}{\sqrt{D}}$. An alternative options for the InfoNCE could be the CLOOB loss (Fürst et al., 2022; Sanchez-Fernandez et al., 2023).

**Training on Ligand-Based Data.** When training on ligand-based datasets, we leverage large collections of annotated active and inactive compounds for a given protein target. Since no structural information on the binding pocket is available in this setting, the VN-EGNN module cannot be meaningfully optimized and is therefore kept frozen during training. For each batch, active and inactive compounds are sampled at a ratio of 1:3, and the model is trained with *sigmoid contrastive loss* (Gutmann & Hyvärinen, 2010; Seidl et al., 2023; Zhai et al., 2023), which uses the cosine similarity tween the whole-protein representation $\boldsymbol{p}$ and the corresponding part of the small molecule embeddings $\boldsymbol{m}_{\text{p}m}$, and the activity labels $\boldsymbol{y}$:

$$\mathcal{L}_{\text{LB}}(\boldsymbol{y}, \boldsymbol{p}, \{\boldsymbol{m}_{\text{p}1}, \ldots, \boldsymbol{m}_{\text{p}M}\}) = \text{NCE}(\boldsymbol{y}, \boldsymbol{p}, \{\boldsymbol{m}_{\text{p}1}, \ldots, \boldsymbol{m}_{\text{p}M}\}) =$$

$$= -\frac{1}{M} \sum_{m=1}^{M} \left( y_m \log(\sigma(s(\boldsymbol{p}, \boldsymbol{m}_{\text{p}m}))) + (1 - y_m) \log(1 - \sigma(s(\boldsymbol{p}, \boldsymbol{m}_{\text{p}m}))) \right), \tag{12}$$

where $y_m \in \{0, 1\}$ denotes the activity label for the protein-ligand pair and $\sigma$ is the sigmoid function.

## 4 EXPERIMENTS AND RESULTS

We train models and evaluate CONGLUDE's performance on four drug-discovery tasks: virtual screening (Section 4.1), binding-pocket prediction (Section 4.3), ligand-conditioned pocket selection (Section 4.4), and target fishing (Section 4.2). The first two are widely studied with established benchmarks, whereas the latter two are more data-poor and have less standardized benchmarks and baselines. Train and test datasets are detailed in Section D, training procedures in Section E, and task-specific metrics in Section F.

### 4.1 VIRTUAL SCREENING

We compare our method with the classical docking methods Surflex-Dock (Spitzer & Jain, 2012), AutoDock Vina (Trott & Olson, 2010) and Glide-SP (Halgren et al., 2004), the machine-learning

Table 1: Zero-shot performance on virtual screening on the DUD-E and LIT-PCBA datasets measured by AUROC, BEDROC and EF at 1%. For ConGLUDE we report the median and mean-absolute-deviation over three training re-runs. Best value per column is marked in bold; values within the MAD of the best are also highlighted.

| | DUD-E | | | LIT-PCBA | | |
|---|---|---|---|---|---|---|
| | AUROC↑ | BEDROC↑ | EF 1% ↑ | AUROC↑ | BEDROC↑ | EF 1% ↑ |
| Surflex-Dock[b] | – | – | – | 51.47 | – | 2.50 |
| AutoDock Vina[b] | 71.60 | – | 7.32 | – | – | – |
| Glide-SP[b] | 76.70 | 40.70 | 16.18 | 53.15 | 4.00 | 3.41 |
| RF-Score[b] | 65.21 | 12.41 | 4.52 | – | – | – |
| NNScore [b] | 68.30 | 12.20 | 4.02 | – | – | – |
| GninA[b] | – | – | – | 60.93 | 5.40 | 4.63 |
| Pafnucy[b] | 63.11 | 16.50 | 3.86 | – | – | – |
| OnionNet[b] | 59.71 | 8.62 | 2.84 | – | – | – |
| DeepDTA[b] | – | – | – | 56.27 | 2.53 | 1.47 |
| BigBind[b] | – | – | – | 60.80 | – | 3.82 |
| PLANET[b] | 71.60 | – | 8.83 | 57.31 | – | 3.87 |
| DrugCLIP[b] | **80.93** | **50.52** | **31.89** | 57.17 | 6.23 | 5.51 |
| SPRINT | 69.01[a] | 13.26[a] | 4.85[a] | **73.40**[c] | **12.30**[c] | **10.78**[c] |
| ConGLUDe (ours) | **81.29** | **49.49** | **31.76** | 64.06 | **12.24** | **11.03** |
| | ±1.11 | ±1.94 | ±1.13 | ±3.25 | ±2.06 | ±1.81 |

[a] evaluated in this work.  [b] values from Gao et al. (2024).  [c] values from McNutt et al. (2024).

based scoring functions RF-Score (Ballester & Mitchell, 2010a), NNScore (Durrant & McCammon, 2011) and GninA (McNutt et al., 2021), deep learning methods predicting pocket-ligand interactions, Pafnucy (Stepniewska-Dziubinska et al., 2017), OnionNet (Zheng et al., 2019), DeepDTA (Öztürk et al., 2018), BigBind (Brocidiacono et al., 2024) and PLANET (Zhang et al., 2024), as well as the contrastive learning-based virtual screening methods DrugCLIP (Gao et al., 2024) and SPRINT (McNutt et al., 2024). Results on AUROC, BEDROC, and enrichment factor at 1% can be found in Table 1, and additional results on EF 0.5% and 5% are shown in Appendix tables G1 and G2. ConGLUDE performs on par with the best method, DrugCLIP, on DUD-E, and for BEDROC and EF 1% metrics is also on par with the best method on LIT-PCBA, which is SPRINT. Notably, ConGLUDE clearly outperforms DrugCLIP on LIT-PCBA and SPRINT on DUD-E, demonstrating strong cross-benchmark generalization.

## 4.2 Zero-Shot Target Fishing

We evaluated ConGLUDe on target fishing data from Reinecke et al. (2024), which contain drug targets for ≈1,000 ligands. The biotechnology to determine drug targets, called Kinobeads chemical-proteomics, is vastly different from the training data of ConGLUDe, and thus the datasets constitutes a challenging new domain, which we approach zero-shot. We encoded each ligand and protein using ConGLUDe, and ranked the potential target proteins for each ligand by the cosine similarity. We compare our results to DrugCLIP (Gao et al., 2024), SPRINT (McNutt et al., 2024), and DiffDock (Corso et al., 2023)(Table 2). Since DrugCLIP requires pocket structures as input, which are not available for the target fishing dataset, we use the pocket prediction methods P2Rank and VN-EGNN to generate candidate pockets. ConGLUDe consistently outperforms all baselines, highlighting its effectiveness for drug target identification. In contrast, all baselines except DiffDock exhibit performance close to random, underscoring the challenge of the task and the advantage of our unified approach.

## 4.3 Binding Site Prediction

As shown in Table 4.3, we retain the performance of VN-EGNN Sestak et al. (2024) on binding site prediction which highlights the robustness of the VN-EGNN encoder. The architectural modifications

Table 2: Zero-shot performance on target fishing task measured by AUROC, $\Delta$AUPRC and EF at 1%. Best value per column is marked in bold.

|  | AUROC ↑ | $\Delta$AUPRC ↑ | EF 1% ↑ |
|---|---|---|---|
| P2Rank+DrugCLIP | $51.18 \pm 17.51$ | $0.56 \pm 2.05$ | $1.36 \pm 6.36$ |
| VN-EGNN+DrugCLIP | $54.21 \pm 20.05$ | $0.36 \pm 1.32$ | $0.83 \pm 7.57$ |
| SPRINT | $42.47 \pm 14.31$ | $0.29 \pm 1.21$ | $0.83 \pm 3.32$ |
| DiffDock | $58.94 \pm 17.73$ | $2.17 \pm 4.51$ | $5.26 \pm 14.05$ |
| ConGLUDe (ours) | $\mathbf{65.63} \pm 20.44$ | $\mathbf{5.10} \pm 10.19$ | $\mathbf{9.88} \pm 19.42$ |

and adaptations to support additional tasks do not hinder its ability to perform well on pocket prediction. Full results of compared methods from Sestak et al. (2024) are in Appendix Table G3.

Table 3: Performance at binding site identification in terms of DCC and DCA success rates on the COACH420, HOLO4K, and PDBbind datasets. Best value marked bold.

| Methods | COACH420 | | HOLO4K | | PDBbind2020 | |
|---|---|---|---|---|---|---|
|  | DCC↑ | DCA↑ | DCC↑ | DCA↑ | DCC↑ | DCA↑ |
| VN-EGNN | **0.605** | **0.750** | **0.532** | 0.659 | 0.669 | 0.820 |
| ConGLUDe (ours) | 0.602 | 0.726 | 0.525 | **0.693** | **0.689** | **0.856** |

## 4.4 LIGAND-CONDITIONED POCKET SELECTION

We also performed *ligand-conditioned pocket selection*, where candidate pockets are ranked by their likelihood to bind to a given ligand, which is in contrast with unconditioned predictors that ignore ligand information. We compared ConGLUDe to a docking-based method (DiffDock (Corso et al., 2023)) and two unconditioned baselines (P2Rank (Krivák & Hoksza, 2018), VN-EGNN (Sestak et al., 2024)). Additionally, we implemented a two-step approach pairing DrugCLIP (Gao et al., 2024) with a pocket predictor (P2Rank or VN-EGNN), ranking candidate pockets by the similarity between the DrugCLIP-encoded pocket and ligand. Unlike docking, which simulates every ligand–pocket pair, ConGLUDe embeds ligands and pockets separately and scores them via a dot product, offering a major speed advantage. We evaluated on a PDBbind time split (Stärk et al., 2022) and the ASD benchmark enriched for allosteric sites (Liu et al., 2020), reporting Top-1 DCC@4Å. ConGLUDe outperforms DiffDock and both unconditioned baselines on PDBbind (Table 4). On ASD, performance drops for all methods due to allosteric pockets that are rarely seen during training, and unconditioned predictors frequently miss these sites. ConGLUDe , unlike DrugCLIP still improves ligand-specific selection over unconditioned baselines, but overall accuracy is limited by VN-EGNN's detection of allosteric pockets. Details in Appendix Section G.3.

Table 4: Performance of ligand-conditioned pocket selection measured by the top-1 DCC success rate at a 4Å threshold. Values in parentheses indicate 95% confidence intervals. The best-performing method is highlighted in bold .

|  | PDBBind Time DCC↑ | ASD DCC↑ |
|---|---|---|
| DiffDock | 0.37 (0.33, 0.42) | **0.33** (0.31,0.35) |
| P2Rank | 0.45 (0.41, 0.50) | 0.24 (0.23, 0.26) |
| P2Rank+DrugCLIP | 0.42 (0.37,0.47) | 0.24 (0.22,0.26) |
| VN-EGNN | 0.39 (0.34,0.43) | 0.20 (0.18,0.21) |
| VN-EGNN+DrugCLIP | 0.41 (0.36,0.45) | 0.19 (0.18,0.21) |
| ConGLUDe (ours) | **0.47** (0.43,0.52) | 0.29 (0.27,0.30) |

## 4.5 ABLATION STUDIES

We performed an ablation study on the main components of ConGLUDe: a) structure-based training data, b) ligand-based training data, c) geometric loss, d) contrastive loss between molecule and protein, and e) contrastive loss between molecule and binding site. The results of the ablation study are shown in Table 5. On LIT-PCBA, ablating each component leads to a deterioration of the performance metrics, which indicates that all components together contribute to the effectiveness of ConGLUDe. On the DUD-E benchmark, structure-based data are critical for performance, while ligand-based data are not strictly necessary and, in fact, removing them even slightly improves results, albeit at the cost of reduced generalizability to more realistic datasets such as LIT-PCBA. Ablating individual loss terms, that are not directly responsible for virtual screening, has little effect on virtual screening performance. Nevertheless, all components are essential for enabling the learning of other tasks, including pocket prediction ($\mathcal{L}_{\text{geometric}}$), pocket selection ($\mathcal{L}_{\text{m2b}}$), and target fishing ($\mathcal{L}_{\text{m2p}}$).

Table 5: Ablation study: Performance on virtual screening on the DUD-E (Mysinger et al., 2012) and LIT-PCBA (Tran-Nguyen et al., 2020) datasets measured by AUROC, BEDROC and EF at 1%. ConGLUDe results are reported from a single run using the same random seed as the ablated models.

| | DUD-E | | | LIT-PCBA | | |
|---|---|---|---|---|---|---|
| | AUROC↑ | BEDROC↑ | EF 1% ↑ | AUROC↑ | BEDROC↑ | EF 1% ↑ |
| only SB data | 83.88 | 56.20 | 36.57 | 53.06 | 5.48 | 4.73 |
| only LB data | 67.11 | 10.61 | 5.31 | 67.94 | 11.11 | 9.38 |
| no $\mathcal{L}_{\text{geometric}}$ | 83.26 | 53.05 | 34.79 | 64.17 | 11.41 | 10.06 |
| no $\mathcal{L}_{\text{m2p}}$ | 82.58 | 50.30 | 32.26 | 64.80 | 11.01 | 10.24 |
| no $\mathcal{L}_{\text{m2b}}$ | 81.55 | 50.16 | 32.34 | 64.90 | 10.97 | 8.98 |
| ConGLUDe | 82.04 | 50.80 | 32.52 | 66.25 | 13.63 | 12.25 |

## 5 CONCLUSION, LIMITATIONS AND DISCUSSION

We introduce ConGLUDe, an approach that combines structure- and ligand-based drug design via an architecture that can profit from both ligand- and structure-based training data. In difficult, zero-shot virtual screening benchmarks, ConGLUDe reaches state-of-the-art, and can also solve multiple other tasks, such as binding pocket identification, ligand-conditioned pocket selection, and target fishing. **Limitations.** Our method can be applied to proteins with experimentally resolved 3D structures as they appear in PDB. Although we performed well in difficult zero-shot settings, it is unclear how the performance changes for predicted 3D structures or proteins that are very distant from any proteins that occur in PDB. Similarly, our method performs well for typical drug-like small molecules and natural ligands, but we have not explored how the performance changes for small molecules from very distant chemical spaces. **Discussion.** Our results indicate that a Deep Learning architecture that effectively uses both structure- and ligand-based data and combines it into a single model, can be considered as a foundation model for drug discovery. Nevertheless, we envision that our paradigm can lead to even more precise and powerful models, perhaps in combination with generative approaches.

## ETHICS STATEMENT

This work relies exclusively on publicly available datasets for computational drug discovery, and no experiments involving humans or animals were conducted.

## REPRODUCIBILITY STATEMENT

All datasets used in this work are public. We will release the complete code for all experiments, including scripts for data download/pre-processing, fixed train/val/test splits, configuration files with exact hyperparameters and random seeds, and evaluation code for the metrics. We will also provide pre-trained checkpoints, as well as an installation guide for the used libraries.

## USE OF LARGE LANGUAGE MODELS

Large Language Models (LLMs) were used in the preparation of this manuscript to improve the grammar, readability, and stylistic consistency of texts written by the authors. LLM-based tools also assisted in literature searches. All scientific concepts, analyses, figures, and results were developed, implemented, and validated solely by the authors. Code development was likewise carried out by the authors, with code-assistance tools (e.g., GitHub Copilot, Claude Code) used only to debug or refine existing implementations and for narrowly defined tasks under explicit author guidance. At no point were LLMs used to generate research ideas or explore scientific concepts.

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

# A NOTATION

The following table summarizes all the notation used throughout this paper.

| Definition | Symbol | Type |
|---|---|---|
| **Scalars** | | |
| batch size | $J$ | $\mathbb{N}$ |
| contrastive space dimension | $D$ | $\mathbb{N}$ |
| VN-EGNN output dimension | $E$ | $\mathbb{N}$ |
| number of binding site VNs | $N$ | $\mathbb{N}$ |
| number of predicted binding sites after clustering | $K$ | $\mathbb{N}$ |
| number of labeled small molecules for a given protein | $M$ | $\mathbb{N}$ |
| number of protein residues | $S$ | $\mathbb{N}$ |
| **Representations** | | |
| protein residue representation | $\boldsymbol{h}'_s$ | $\mathbb{R}^E$ |
| pocket representation before clustering | $\boldsymbol{b}'_n$ | $\mathbb{R}^E$ |
| protein representation before projection | $\boldsymbol{p}'$ | $\mathbb{R}^E$ |
| pocket representation before projection | $\hat{\boldsymbol{b}}_k$ | $\mathbb{R}^E$ |
| final protein representation | $\boldsymbol{p}$ | $\mathbb{R}^D$ |
| final pocket representation | $\boldsymbol{b}_k$ | $\mathbb{R}^D$ |
| small molecule representation | $\boldsymbol{m}_m = [\boldsymbol{m}_{\mathrm{p}m}, \boldsymbol{m}_{\mathrm{b}m}]$ | $\mathbb{R}^{2D}$ |
| **Coordinates** | | |
| protein residue position | $\mathbf{x}'_s$ | $\mathbb{R}^3$ |
| pocket node position before clustering | $\mathbf{z}'_k$ | $\mathbb{R}^3$ |
| predicted binding pocket center/final VN position | $\hat{\mathbf{z}}_n$ | $\mathbb{R}^3$ |
| **Data quantities** | | |
| predicted confidence value for $\hat{\mathbf{z}}_n$ | $\hat{c}_n$ | $\mathbb{R}$ |
| ground-truth confidence value for $\hat{\mathbf{z}}_n$ | $c_n$ | $\{c_0, [0.5, 1]\}$ |
| residue-level binding site label | $z_s$ | $\{0, 1\}$ |
| binary activity label for molecule $\boldsymbol{m}_m$ | $y_m$ | $\{0, 1\}$ |
| **Constants** | | |
| fall-back value for confidence calculation | $c_0$ | 0.001 |
| tolerance radius for confidence calculation | $\gamma$ | 4.0 |
| temperature for $\mathcal{L}_{\mathrm{p2m}}$ | $\tau_{\mathrm{p2m}}$ | $\frac{1}{\sqrt{2D}}$ |
| temperature for $\mathcal{L}_{\mathrm{m2p}}$ | $\tau_{\mathrm{m2p}}$ | $\frac{1}{\sqrt{D}}$ |
| temperature for $\mathcal{L}_{\mathrm{m2b}}$ | $\tau_{\mathrm{m2b}}$ | $\frac{1}{\sqrt{D}}$ |
| **Functions** | | |
| cosine similarity | $s(.,.)$ | $\mathbb{R}^D \times \mathbb{R}^D \to [-1, 1]$ |
| sigmoid function | $\sigma(.,.)$ | $\mathbb{R} \to [0, 1]$ |

The InfoNCE loss used for structure-based training is defined as follows:

$$\mathrm{InfoNCE}(\boldsymbol{q}^{(j)}, \boldsymbol{k}^{(j)}, \{\boldsymbol{k}^{(1)}, \ldots, \boldsymbol{k}^{(J)}\}; \tau) = -\log \frac{\exp\left(s(\boldsymbol{q}^{(j)}, \boldsymbol{k}^{(j)})/\tau\right)}{\sum_{i=1}^{J} \exp\left(s(\boldsymbol{q}^{(j)}, \boldsymbol{k}^{(i)})/\tau\right)}. \tag{A.1}$$

# B COMPARED METHODS

Virtual screening methods can broadly be classified into two families: physics- and knowledge-driven docking engines that search conformational space and apply handcrafted or empirical scoring, and machine-learning scoring functions that learn structure–activity relationships from data. Classical docking methods such as Glide-SP (Halgren et al., 2004), AutoDock Vina (Trott & Olson, 2010), and Surflex (Spitzer & Jain, 2012) generate ligand poses within a predefined protein pocket and rank them using empirical scoring functions that combine physics-inspired energy terms. Building on these, pose-based machine learning methods like NN-Score (Durrant & McCammon, 2011), RF-Score (Ballester & Mitchell, 2010a), and the CNN-augmented docking framework Gnina (McNutt et al., 2021) operate on already docked complexes, predicting binding affinity or pose quality from structural features of the protein–ligand arrangement.

To move beyond handcrafted features, a series of deep learning models have been proposed that also require an explicit binding pocket. Examples include 3D CNNs such as Pafnucy (Stepniewska-Dziubinska et al., 2017), which voxelize the local binding site; distance-shell descriptors as in OnionNet (Zheng et al., 2019); and graph neural networks approaches like BigBind (Brocidiacono et al., 2024) and PLANET (Zhang et al., 2024). These methods explicitly exploit geometric and chemical details of the binding environment and generally aim to rescore or refine docking outputs.

More recently, contrastive learning approaches have been introduced to bridge proteins and ligands directly. DrugCLIP (Gao et al., 2024) learns joint representations by contrasting ligands with explicit binding pocket embeddings, while SPRINT (McNutt et al., 2024) adopts a sequence-based whole-protein representation to align ligands with their corresponding targets.

## C  VN-EGNN Details

### C.1  Heterogeneous Message Passing

Following Sestak et al. (2024), we briefly summarize the heterogeneous message passing scheme used in VN-EGNN. Each layer consists of three message passing steps that exchange information between protein residues ($\mathcal{R}$) and virtual binding pocket nodes ($\mathcal{B}$).

The first step corresponds to the standard equivariant graph neural network (EGNN) formulation (Satorras et al., 2021), where information is exchanged between neighboring protein residues:

**Message passing step 1 ($\mathcal{R} \to \mathcal{R}$):**

$$\boldsymbol{\mu}_{ij}^{(\mathcal{RR})} = \boldsymbol{\phi}_{e^{(\mathcal{RR})}}(\boldsymbol{h}_i, \boldsymbol{h}_j, \|\mathbf{x}_i - \mathbf{x}_j\|) \tag{C.1}$$

$$\boldsymbol{\mu}_i^{(\mathcal{RR})} = \frac{1}{|\mathcal{N}(i)|} \sum_{j \in \mathcal{N}(i)} \boldsymbol{\mu}_{ij}^{(\mathcal{RR})} \tag{C.2}$$

$$\mathbf{x}_i = \mathbf{x}_i + \frac{1}{|\mathcal{N}(i)|} \sum_{j \in \mathcal{N}(i)} \frac{\mathbf{x}_i - \mathbf{x}_j}{\|\mathbf{x}_i - \mathbf{x}_j\|} \phi_{\mathbf{x}^{(\mathcal{RR})}}(\boldsymbol{\mu}_{ij}^{(\mathcal{RR})}) \tag{C.3}$$

$$\boldsymbol{h}_i = \boldsymbol{h}_i + \phi_{h^{(\mathcal{RR})}}\left(\boldsymbol{h}_i, \boldsymbol{\mu}_i^{(\mathcal{RR})}\right). \tag{C.4}$$

Here, the coordinates $\mathbf{x}_i$ and features $\boldsymbol{h}_i$ of residue nodes are updated based on aggregated messages from their neighbors. The MLPs $\phi_{e^{(\mathcal{RR})}}$, $\phi_{\mathbf{x}^{(\mathcal{RR})}}$, and $\phi_{h^{(\mathcal{RR})}}$ are learnable functions specific to each layer. The same applies to all MLPs $\phi$. in the subsequent steps.

In the second step, residue nodes transmit information to virtual pocket nodes $\mathcal{B}$, which act as proxies for potential binding sites:

**Message passing step 2 ($\mathcal{R} \to \mathcal{B}$):**

$$\boldsymbol{\mu}_{ij}^{(\mathcal{RB})} = \boldsymbol{\phi}_{e^{(\mathcal{RB})}}(\boldsymbol{b}_i, \boldsymbol{h}_j, \|\mathbf{z}_i - \mathbf{x}_j\|) \tag{C.5}$$

$$\boldsymbol{\mu}_i^{(\mathcal{RB})} = \frac{1}{S} \sum_{j=1}^{S} \boldsymbol{\mu}_{ij}^{(\mathcal{RB})} \tag{C.6}$$

$$\mathbf{z}_i = \mathbf{z}_i + \frac{1}{S} \sum_{j=1}^{S} \frac{\mathbf{z}_i - \mathbf{x}_j}{\|\mathbf{z}_i - \mathbf{x}_j\|} \phi_{\mathbf{x}^{(\mathcal{RB})}}(\boldsymbol{\mu}_{ij}^{(\mathcal{RB})}) \tag{C.7}$$

$$\boldsymbol{b}_i = \boldsymbol{b}_i + \phi_{h^{(\mathcal{RB})}}\left(\boldsymbol{b}_i, \boldsymbol{\mu}_i^{(\mathcal{RB})}\right) \tag{C.8}$$

Finally, the third step propagates information in the reverse direction, from virtual nodes back to residue nodes:

**Message passing step 3 ($\mathcal{B} \to \mathcal{R}$):**

$$\boldsymbol{\mu}_{ij}^{(\mathcal{BR})} = \phi_{e^{(\mathcal{BR})}}(\boldsymbol{h}_i, \boldsymbol{b}_j, \|\mathbf{x}_i - \mathbf{z}_j\|) \tag{C.9}$$

$$\boldsymbol{\mu}_i^{(\mathcal{BR})} = \frac{1}{N} \sum_{j=1}^{N} \boldsymbol{\mu}_{ij}^{(\mathcal{BR})} \tag{C.10}$$

$$\mathbf{x}_i = \mathbf{x}_i + \frac{1}{N} \sum_{j=1}^{N} \frac{\mathbf{x}_i - \mathbf{z}_j}{\|\mathbf{x}_i - \mathbf{z}_j\|} \phi_{\mathbf{x}^{(\mathcal{BR})}}(\boldsymbol{\mu}_{ij}^{(\mathcal{BR})}) \tag{C.11}$$

$$\boldsymbol{h}_i = \boldsymbol{h}_i + \phi_{h^{(\mathcal{BR})}}\left(\boldsymbol{h}_i, \boldsymbol{\mu}_i^{(\mathcal{BR})}\right) \tag{C.12}$$

## C.2 OBJECTIVE FUNCTIONS

VNEGNN (Sestak et al., 2024) is trained using a combination of losses that supervise the prediction of binding site centers, residue-level segmentation, and confidence of the predictions.

To ensure accurate prediction of the binding site center (bsc) location, the squared distance between the true binding site center $\mathbf{z}$ and the closest predicted center $\hat{\mathbf{z}}_n$ among $N$ candidates is minimized:

$$\mathcal{L}_{\text{bsc}}(\{\hat{\mathbf{z}}_1, \dots, \hat{\mathbf{z}}_N\}, \mathbf{z}) = \min_{n \in 1, \dots, N} \|\mathbf{z} - \hat{\mathbf{z}}_n\|^2. \tag{C.13}$$

For residue-level binding site segmentation, the network outputs predictions for each residue $s$ through a multilayer perceptron: $\hat{z}_s = MLP(\boldsymbol{h}_s')$. The segmentation loss is defined as a differentiable Dice loss, which compares the predicted and true residue labels $z_s$:

$$\mathcal{L}_{\text{seg}}(\{\hat{z}_1, \dots, \hat{z}_S\}, \{z_1, \dots, z_S\}; \epsilon) = 1 - \frac{2 \sum_{s=1}^{S} z_s \hat{z}_s + \epsilon}{\sum_{n=1}^{N} z_s + \sum_{n=s}^{S} \hat{z}_s + \epsilon}, \tag{C.14}$$

where $\epsilon$ is a small constant to stabilize the division.

Moreover, each predicted center $\hat{\mathbf{z}}_n$ is assigned a confidence score $\hat{c}_n$ which should reflect its proximity to the true center. The target confidence $c_n$ is defined as:

$$c_n = \begin{cases} 1 - \frac{1}{2\gamma} \cdot \|\mathbf{z} - \hat{\mathbf{z}}_n\| & \text{if } \|\mathbf{z} - \hat{\mathbf{z}}_n\| \leqslant \gamma, \\ c_0 & \text{otherwise,} \end{cases}, \tag{C.15}$$

and the corresponding confidence loss is the mean squared error between predicted and target confidences:

$$\mathcal{L}_{\text{confidence}}(\{\hat{c}_1, \dots, \hat{c}_N\}, \{c_1, \dots, c_N\}) = \frac{1}{N} \sum_{n=1}^{N} (c_n - \hat{c}_n)^2. \tag{C.16}$$

The total VNEGNN objective combines the three components and is used as the geometric learning objective in ConGLUDe's structure-based training:

$$\mathcal{L}_{\text{geometric}} = \mathcal{L}_{\text{bsc}} + \mathcal{L}_{\text{seg}} + \mathcal{L}_{\text{confidence}}. \tag{C.17}$$

## D DATASETS

### D.1 STRUCTURE-BASED TRAINING DATASETS

For structure-based training, we utilized subsets of PDBBind v.2020 (Wang et al., 2005), adopting the dataset partitions established by the baseline methods corresponding to each task. Specifically,

for virtual screening, we followed the DrugCLIP split (Gao et al., 2024). For binding site prediction, we trained on scPDB, consistent with VN-EGNN (Sestak et al., 2024), and for ligand-conditioned pocket selection, we employed the time-based split used in DiffDock (Corso et al., 2023).

## D.2 Ligand-Based Training Datasets

For the training on the ligand-based data, we employed the MERGED dataset introduced in SPRINT (McNutt et al., 2024), which integrates data from PubChem (Kim et al., 2024), BindingDB (Gilson et al., 2015), and ChEMBL (Gaulton et al., 2011). We use the combined MERGED training and test splits as the basis for our training set and keep the same validation split as (McNutt et al., 2024). For each protein in the dataset, we used an available 3D structure from the PDB; if none was available, we generated an AlphaFold2 structure. To prevent information leakage, proteins with more than 90% sequence identity to any test protein were excluded, using MMSeqs2 (Steinegger & Söding, 2017) with a coverage threshold of 0.8. The number of unique proteins and total data points for each task subset can be found in Table D1.

Table D1: Number of PLI data points in structure-based (SB) and ligand-based (LB) training and validation datasets.

| | SB Data | | LB Data | | | |
| | Train | Val | Train | | Val | |
| Task | Complexes | | Proteins | Data Points | Proteins | Data Points |
| --- | --- | --- | --- | --- | --- | --- |
| Virtual Screening | 24,896 | 400 | 3,526 | 56,187,278 | 47 | 5,809,414 |
| Pocket Prediction | 14,564 | 1,610 | 3,103 | 49,493,389 | 44 | 5,539,515 |
| Pocket Selection | 24,127 | 1,384 | 3,685 | 57,096,449 | 45 | 5,523,271 |

## D.3 Test Datasets

We evaluated our models on diverse benchmark datasets tailored to each task.

For virtual screening, we used two widely adopted benchmarks, DUD-E (Mysinger et al., 2012) and LIT-PCBA (Tran-Nguyen et al., 2020). The DUD-E dataset contains 22,886 active compounds against 102 protein targets, paired with property-matched decoys designed to mimic physical characteristics of active molecules while differing in topology. LIT-PCBA complements DUD-E by providing experimentally validated high-throughput screening results across 15 targets. Unlike DUD-E, which uses synthetic decoys, LIT-PCBA relies exclusively on assay data, resulting in a more realistic and more challenging benchmark for large-scale virtual screening.

Pocket prediction performance was evaluated on three established datasets, which were also used in Sestak et al. (2024). Coach420 (Krivák & Hoksza, 2018) is a curated benchmark of 420 proteins with annotated binding sites on single-chain structures. HOLO4K (Krivák & Hoksza, 2018) consists of over 4,000 holo protein structures with experimentally verified binding pockets, many of which are large multi-chain complexes. For both, Coach420 and HOLO4K, we adopt the so-called mlig subsets, as detailed in Krivák & Hoksza (2018), which encompass only biologically relevant ligands. Finally, the PDBBind v.2020 refined set (Wang et al., 2005) includes high-quality protein–ligand complexes with reliable structural and binding affinity data, serving as a stringent benchmark for pocket localization in realistic docking scenarios.

For ligand-conditioned pocket selection, we employed the temporal test split of PDBBind introduced in EquiBind (Stärk et al., 2022), which ensures temporal separation between training and evaluation complexes, thereby simulating prospective prediction performance. In addition, we constructed a new benchmark based on the Allosteric Site Database (ASD, June 2023 release) (Liu et al., 2020). This dataset comprises protein–ligand complexes annotated with allosteric binding sites, providing a novel and challenging testbed for evaluating the generalization of models beyond orthosteric binding interactions. We filtered out all proteins overlapping with the PDBbind training and validation sets proteins.

For target fishing, we use the Kinobeads chemical-proteomics dataset of Reinecke et al. (2024). The study profiled 1,183 kinase-directed small molecules in cancer-cell lysates by competitive enrichment

Table D2: Summary of test datasets used for evaluation across different tasks. LB = ligand-based datasets, SB = structure-based datasets.

| Dataset | Type | Data Points | Unique Proteins | Unique Ligands |
|---|---|---|---|---|
| DUD-E | LB | 1,434,019 | 102 | 1,200,431 |
| LIT-PCBA | LB | 2,808,770 | 15 (129) | 383,772 |
| Coach420 | SB | 348 | 300 | 278 |
| HOLO4K | SB | 4235 | 3,446 | 1,700 |
| PDBbind Refined | SB | 5,309 | 5,309 | 4,482 |
| ASD | SB | 1802 | 1765 | 1117 |
| PDBbind Time | SB | 384 | 321 | 328 |
| Kinobeads | LB | 1,424,686 | 2,714 | 985 |

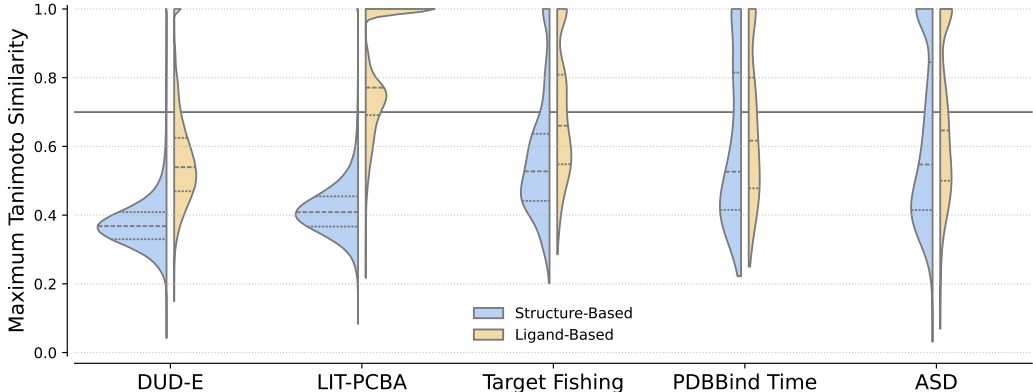

Figure D1: Distributions of maximum Tanimoto similarities between ECFP count fingerprints (radius 2, length 2048) of test-set molecules and those in the structure- and ligand-based training sets.

on immobilized inhibitors, and reports apparent affinities (Kd$^{app}$) from a two-dose competition design (100 nM and 1 μM) and provides high-confidence target calls via a trained random-forest classifier. We treat these calls as positives and use the remaining measured proteins as negatives when ranking targets per compound. The raw data are publicly available via ProteomicsDB. After pre-processing and mapping gene symbols to the PDB structure with the highest resolution among those annotated for Homo sapiens (if available), we obtained a dataset of 985 ligands and 2,714 proteins.

Table D2 summarizes the number of data points, unique proteins and unique ligands for each test dataset. Figure D1 shows the distribution of maximum ECFP Tanimoto similarities between each test molecules and all training molecules.

## E    HYPERPARAMETERS AND TRAINING DETAILS

For the protein encoder, we adopt VN-EGNN with the default parameters reported by Sestak et al. (2024), i.e., a 5-layer architecture with distinct weights per layer, input dimension 1280 (from ESM-2 embeddings (Lin et al., 2023)), output dimension 100, SiLU activation, and residual connections. Two linear projection layers are trained to map binding site and protein nodes into the contrastive space of dimension $D = 256$.

Ligands are represented as extended connectivity fingerprints (ECFP6)(Rogers & Hahn, 2010) of length 2048, concatenated with a vector of 210 chemical descriptors from RdKit (Landrum & contributors, 2006), yielding an input dimension of 2258. The ligand encoder is a two-layer MLP with hidden dimension 512, output dimension $2D = 512$, GELU activation, 10% input dropout, and 50% dropout on the hidden layer.

Training uses a batch size of 64 on structure-based data, resulting in 63 negative ligands per protein and vice versa through in-batch negative sampling. For ligand-based training, each batch contains 16 proteins, with actives and inactives sampled at a 1:3 ratio and capped at 10,000 active ligands per protein. Contrastive loss temperature parameters are set to the inverse square root of the respective embedding dimensions, i.e., $\tau_{\mathrm{p2m}} = \frac{1}{\sqrt{2D}}$ and $\tau_{\mathrm{m2p}} = \tau_{\mathrm{m2b}} = \frac{1}{\sqrt{D}}$. All loss terms are weighted equally in structure-based training, while the ligand-based loss is scaled by a factor of 6 to match the magnitude of $L_{SB}$.

We optimize using AdamW (Loshchilov & Hutter, 2019) with an initial learning rate of $10^{-3}$. A learning rate scheduler reduces the rate by a factor of 10 when the validation metric does not improve for 30 epochs, with a minimum learning rate of $10^{-6}$. Early stopping with a patience of 100 epochs is applied based on the same validation metric. Separate models were trained for each task due to the different data splits and training was conducted on NVIDIA A100 GPUs with 40GB memory for 200–350 epochs.

## F METRICS

Depending on the task, we employ different evaluation metrics, which are formally described below.

For virtual screening, we evaluate the area under the receiver operating characteristic curve (AUROC), the Boltzmann-enhanced discrimination of ROC (BEDROC) at $\alpha = 85$, and enrichment factors (EF) at different top 0.5%, 1% and 5%.

Unlike AUROC, which treats all parts of the ranking equally and is therefore a strong general-purpose metric, BEDROC is tailored to virtual screening scenarios where early recognition of actives is critical (Truchon & Bayly, 2007). The enrichment factor at top $x\%$ quantifies the overrepresentation of actives among the highest-ranked molecules. An EF of 1 corresponds to random ranking, while larger values indicate stronger enrichment.

For binding pocket prediction, we measure the DCC (distance from predicted pocket center to ground-truth pocket center) or DCA (distance from predicted pocket center to the closest atom of the corresponding ligand) success rates at 4 Å. For a protein with $k$ ground-truth pockets, we consider the $k$ top-ranked binding sites. The success rate is the fraction of ground-truth pockets where at least one predicted pocket satisfies the DCC or DCA threshold of 4 Å.

For ligand-conditioned pocket selection, we consider the DCC success rate of the top-ranked predicted pocket compared to all ground-truth pockets associated with the given ligand.

## G EXTENDED RESULTS

### G.1 VIRTUAL SCREENING

Tables G1 and G2 show the complete evaluation on DUD-E and LIT-PCBA split by dataset. As an additional evaluation, we report DrugCLIP (Gao et al., 2024) results using predicted binding pockets from P2Rank (Krivák & Hoksza, 2018) or VN-EGNN (Sestak et al., 2024), simulating a setting where binding pocket information for the target proteins is unavailable. In this scenario, DrugCLIP's performance drops dramatically, highlighting its reliance on prior knowledge of the binding pocket, which is often unavailable in realistic applications.

To visualize the learned representation space, we applied t-SNE to project both protein and ligand embeddings into two dimensions. As shown by one example in Figure G1, active ligands around the embedding of their target protein, whereas inactive ligands are distributed more diffusely across the space. This pattern highlights the model's ability to capture meaningful protein–ligand relationships.

### G.2 BINDING SITE PREDICTION

Table G3 reports performance metrics at binding site identification for different methods similar to Sestak et al. (2024).

Table G1: Zero-shot performance on virtual screening on the LIT-PCBA dataset measured by AUROC, BEDROC and EF at 0.05%, 1% and 5%. For ConGLUDE we report the median and mean-absolute-deviation over three training re-runs. Best value per column is marked in bold; values within the MAD of the best are also highlighted.

| | AUROC (%) | BEDROC (%) | EF | | |
| | | | 0.5% | 1% | 5% |
| --- | --- | --- | --- | --- | --- |
| Surflex | 51.47 | - | - | 2.50 | - |
| Glide-SP | 53.15 | 4.00 | 3.17 | 3.41 | 2.01 |
| Planet | 57.31 | - | 4.64 | 3.87 | 2.43 |
| GninA | 60.93 | 5.40 | - | 4.63 | - |
| DeepDTA | 56.27 | 2.53 | - | 1.47 | - |
| BigBind | 60.80 | - | - | 3.82 | - |
| DrugCLIP | 57.17 | 6.23 | 8.56 | 5.51 | 2.27 |
| P2Rank+DrugCLIP | 49.72 | 2.96 | 2.41 | 2.44 | 1.36 |
| VN-EGNN+DrugCLIP | 52.52 | 3.56 | 1.82 | 2.58 | 1.59 |
| SPRINT | **73.40** | **12.30** | **15.90** | **10.78** | **5.29** |
| ConGLUDe (ours) | 64.06 ±(3.25) | **12.24** ±(2.06) | **15.87** ±(2.03) | 11.03 ±(1.81) | **4.68** ±(0.30) |

Table G2: Zero-shot performance on virtual screening on the DUD-E dataset measured by AUROC, BEDROC and EF at 0.05%, 1% and 5%. For ConGLUDE we report the median and mean-absolute-deviation over three training re-runs. Best value per column is marked in bold; values within the MAD of the best are also highlighted.

| | AUROC (%) | BEDROC (%) | EF | | |
| | | | 0.5% | 1% | 5% |
| --- | --- | --- | --- | --- | --- |
| Glide-SP | 76.70 | 40.70 | 19.39 | 16.18 | 7.23 |
| Vina | 71.60 | - | 9.13 | 7.32 | 4.44 |
| NN-score | 68.30 | 12.20 | 4.16 | 4.02 | 3.12 |
| RFscore | 65.21 | 12.41 | 4.90 | 4.52 | 2.98 |
| Pafnucy | 63.11 | 16.50 | 4.24 | 3.86 | 3.76 |
| OnionNet | 59.71 | 8.62 | 2.84 | 2.84 | 2.20 |
| Planet | 71.60 | - | 10.23 | 8.83 | 5.40 |
| DrugCLIP | 80.93 | **50.52** | 38.07 | **31.89** | **10.66** |
| P2Rank+DrugCLIP | 58.29 | 7.04 | 4.02 | 3.75 | 2.38 |
| VN-EGNN+DrugCLIP | 69.24 | 28.18 | 20.45 | 17.02 | 6.83 |
| ConGLUDe (ours) | **81.29** ±(1.11) | 49.49 ±(1.94) | **39.43** ±(0.97) | 31.76 ±(1.13) | **10.71** ±(0.26) |

### G.3 LIGAND-CONDITIONED POCKET SELECTION

We performed *ligand-conditioned pocket selection*, for which, given a protein structure and a ligand, methods have to rank binding pockets by their likelihood to bind the ligand. Unlike unconditioned pocket predictors that do not have a query ligand as input, our task explicitly conditions on ligand identity and thus supports ligand-specific pocket selection in virtual screening. This is the task that also *blind docking methods* can perform. We compared the following methods (i) DiffDock used as a docking-based selector , and (ii) two unconditioned pocket predictors, P2Rank and VN-EGNN, which always return the same top pocket for a protein regardless of the ligand, and (iii) **ConGLUDe**. Our model embeds the ligand and each candidate pocket and scores their compatibility with a single dot product, which makes inference extremely fast. With precomputed pocket representations, thousands of ligands can be encoded in seconds and scored via dot products. In contrast, docking-based baselines must dock every ligand into every candidate pocket, which is orders of magnitude slower. We evaluated on a PDBbind time-split to assess generalization to future complexes, and the ASD

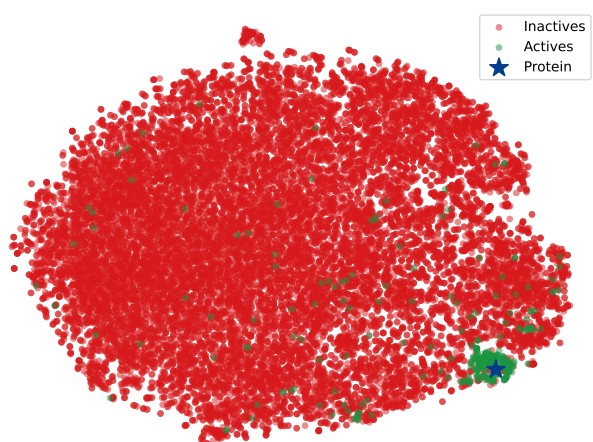

Figure G1: t-SNE projection of protein and ligand embeddings for the DUD-E target with PDB ID 2FSZ.

Table G3: Performance at binding site identification in terms of DCC and DCA success rates. The first column provides the method, the second the number of parameters of the model, the fourth and the fifth column the performance on the COACH420 dataset, the sixth and seventh column the performance on the HOLO4K dataset, and the remaining columns the performance on PDBbind2020. The best performing method(s) per column are marked bold. The second best in italics.

| Methods | COACH420 | | HOLO4K | | PDBbind2020 | |
|---|---|---|---|---|---|---|
| | DCC↑ | DCA↑ | DCC↑ | DCA↑ | DCC↑ | DCA↑ |
| Fpocket | 0.228 | 0.444 | 0.192 | 0.457 | 0.253 | 0.371 |
| P2Rank | 0.464 | *0.728* | 0.474 | **0.787** | 0.653 | 0.826 |
| DeepSite | – | 0.564 | – | 0.456 | – | – |
| Kalasanty | 0.335 | 0.636 | 0.244 | 0.515 | 0.416 | 0.625 |
| DeepSurf | 0.386 | 0.658 | 0.289 | 0.635 | 0.510 | 0.708 |
| DeepPocket | 0.399 | 0.645 | 0.456 | 0.734 | 0.644 | 0.813 |
| GAT | 0.039 | 0.130 | 0.036 | 0.110 | 0.032 | 0.088 |
| GCN | 0.049 | 0.139 | 0.044 | 0.174 | 0.018 | 0.070 |
| GAT + GCN | 0.036 | 0.131 | 0.042 | 0.152 | 0.022 | 0.074 |
| GCN2 | 0.042 | 0.131 | 0.051 | 0.163 | 0.023 | 0.089 |
| SchNet | 0.168 | 0.444 | 0.192 | 0.501 | 0.263 | 0.457 |
| EGNN | 0.156 | 0.361 | 0.127 | 0.406 | 0.143 | 0.302 |
| EquiPocket | 0.423 | 0.656 | 0.337 | 0.662 | 0.545 | 0.721 |
| VN-EGNN | **0.605** | **0.750** | **0.532** | 0.659 | *0.669* | 0.820 |
| ConGLUDe | *0.602* | 0.726 | *0.525* | *0.693* | **0.689** | **0.856** |

benchmark containing allosteric sites and ligands. Candidate pockets are generated once per protein; at test time we rank pockets per ligand. We report top-1 DCC success rate at 4Å On the PDBbind time split, ConGLUDe outperforms both the docking method (DiffDock) and unconditioned pocket prediction baselines (see Table 4). On ASD, overall accuracy is lower for all methods due to the prevalence of allosteric sites that rarely appear in training. Unconditioned predictors, including VN-EGNN, often miss these pockets. Nevertheless, our contrastive ligand–pocket module improves selection of the correct allosteric pocket for a given ligand more often than unconditioned baselines (Table 4). However, the performance of ConGLUDe is limited by the weakness of VN-EGNN at detecting allosteric sites. With an improved detector of allosteric binding pockets, ConGLUDe would also improve. We discuss this also in Limitations.

