# OpenReview forum: "Unifying Structure- and Ligand-based Drug Design via Contrastive Geometric Learning"
_ICLR.cc/2026/Conference — Submitted to ICLR 2026_

### Official Review · Reviewer_ZA3q · 2025-10-28

**Soundness:** 2
**Presentation:** 2
**Contribution:** 2
**Rating:** 4
**Confidence:** 3

**Summary:**

This paper introduces ConGLUDe, a unified framework for drug discovery that jointly leverages both structure-based and ligand-based information through contrastive geometric learning. The architecture integrates a geometric protein encoder (based on VN-EGNN) and a ligand encoder, trained via contrastive learning on structure complexes from pdbbind and ligand data from ChEMBL, PubChem, and BindingDB. ConGLUDe is evaluated on four tasks: virtual screening, binding site prediction, ligand-conditioned pocket selection, and zero-shot target fishing. The method achieves good performance those tasks.

**Strengths:**

1. Different types of tasks are conducted, including virutal screening, binding site prediction, ligand conditioned pocket selection, and target fishing.
2. The improvements on LIT-PCBA dataset is impressive

**Weaknesses:**

1. The paper feels somewhat disjointed, lacking a clear focus. It covers many tasks, but the analysis is not sufficiently deep. Important components like the ablation study are placed in the appendix, and the main text lacks further analysis and visualization.
2. The technical contribution is relatively limited—the work largely appears to be an application of VN-EGNN across different tasks.
3. The performance gains on several tasks are marginal, such as on DUD-E for virtual screening, and the LIGAND-CONDITIONED POCKET SELECTION task in Table 3.

**Questions:**

see Weaknesses

---

> ### Author Response · Authors · 2025-11-28
> **Response to Reviewer ZA3q**
>
> We thank the reviewer for the thoughtful feedback and for recognizing the broad applicability of our method as well as the strong results on LIT-PCBA. Below, we address all raised concerns in detail.
>
> **Weaknesses**
>
> 1. Insufficient analysis:
>
> We agree that parts of our original analysis were too brief or absent, largely due to the number of tasks covered and the strict page limit. With the additional page, we have now moved the ablation study into the main paper and expanded the analysis across all experimental results, including a more thorough discussion of the ablation findings.
>
> 2. Technical contribution:
>
> We disagree with the characterization that this is “an application of VN-EGNN across different tasks,” since VN-EGNN is solely a pocket prediction and encoding module and cannot handle any task that requires ligand information. We agree that our model builds on established components (such as VN-EGNN), but as detailed in our general response, the way these components are integrated differs fundamentally from prior work. Critically, our architectural design *enables combined training* on both structure-based and ligand-based data while still modeling pocket-level interactions. In contrast, existing approaches either rely on explicit pocket information and thus operate only on structure-based data or, when trained on ligand-based data, are unable to provide pocket-specific insights. For completeness and context, we refer the reviewer to the general response, where these points are discussed in more detail.
>
> 3. Performance gains:
>
> While the gains on individual benchmarks may appear modest, as we highlight in the general response, ConGLUDe is the only method to achieve strong, consistent performance across *both* DUD-E and LIT-PCBA. We clearly outperform DrugCLIP on LIT-PCBA and SPRINT on DUD-E. This demonstrates that our unified approach generalizes more robustly across heterogeneous tasks than methods specialized for only one data modality.
>
> In addition, ConGLUDe is the only approach capable of performing ligand-conditioned pocket selection without relying on expensive docking or co-folding, while being competitive with such approaches, specifically DiffDock. We further included comparisons where DrugCLIP is paired with a pocket predictor (P2Rank or VN-EGNN), and find that ranking candidate pockets via DrugCLIP’s ligand-pocket similarity does not improve over the pocket prediction baselines alone.
>
> Finally, in the expanded target-fishing evaluation, ConGLUDe achieves the strongest performance among all compared methods by a substantial margin.
>
> We hope our clarifications and additional analyses fully address the reviewer’s concerns, and we kindly ask the reviewer to consider these revisions when updating their assessment.

---

### Official Review · Reviewer_FPHC · 2025-11-01

**Soundness:** 3
**Presentation:** 3
**Contribution:** 3
**Rating:** 6
**Confidence:** 3

**Summary:**

This paper introduces ConGLUDe framework, which aims to unify structure-based and ligand-based drug design through contrastive geometric learning. ConGLUDe combines geometric protein encoders and ligand encoders, and is jointly trained on multi-source structural and bioactivity data. This unified model can handle multiple drug discovery tasks, achieving SOTA performance and marking a step towards a foundational model for drug discovery.

**Strengths:**

1. This work has interesting motivation, proposing a novel unified framework with very broad applications.

2. The writing is clear and well-structured, with a straightforward model that is easy to understand and follow.

3. The experimental results are comprehensive, demonstrating strong multi-task and zero-shot generalization capabilities.

**Weaknesses:**

1. The loss term calculations ($\mathcal{L}\_{m2p}$ and $\mathcal{L}\_{p2m}$) in Eqn. 9-11 do not align with Figure 2. In Figure 2, the illustrations and descriptions of $\mathcal{L}\_{m2p}$ and $\mathcal{L}\_{p2m}$ appear to be reversed.

2. The current model integrates two data types through alternating training across different batches, which is similar to multi-task learning rather than deep information fusion. Could there be a more intrinsic fusion mechanism? For example, can knowledge learned from large-scale ligand data (such as target selectivity) more directly guide or constrain the learning process from structural data?

3. The discussion of ablation results is insufficient. The ablation experiments in the appendix (Table G4) show that on the DUD-E dataset, the structure-only model significantly outperforms the complete model. While the authors note that DUD-E is less realistic than LIT-PCBA, this result somewhat undermines the core argument that "fusing both data types is crucial." Consider moving this discussion to the main text with a deeper analysis of why this phenomenon occurs.

Minor issues:

a. Many equations lack proper punctuation.

b. Why is there no experimental results analysis in Section 4.2 BINDING SITE PREDICTION?

c. What is the bolding criterion in Table 1? Why do two values appear bolded in the same column? Many appendix tables have similar issues.

**Questions:**

See Weaknesses

---

> ### Author Response · Authors · 2025-11-28
> **Response to Reviewer FPHC**
>
> We thank the reviewer for the positive feedback and for carefully reading our paper. We’re happy that the reviewer appreciated our motivation, the paper’s clarity, and our experimental evaluation. The weaknesses and questions raised are addressed in the following.
>
> **Weaknesses**
>
> 1. Loss terms in Figure 2:
>
> Thanks for spotting this - that was a mistake in the illustration and is now fixed.
>
> 2. Integration of data types:
>
> Great point by the reviewer! While our current training scheme indeed alternates between batches from ligand-based and structure-based datasets, it differs in an important way from standard multi-task learning. In classical multi-task setups, all tasks typically share the same input domain but have different output heads. In our case, however, the model processes two *different* input domains (protein–ligand complexes with explicit 3D geometry vs. large-scale ligand–protein interaction data without structures), and the shared architecture ties these domains together through joint loss terms that are only defined when both representations are available.
>
> Concretely, the same protein and ligand encoders are updated by both structure-based and ligand-based supervision, and the contrastive objectives are defined over their joint representation space. This coupling means that knowledge learned from ligand-based data, e.g., target selectivity patterns, can influence how the protein encoder organizes pockets, and vice versa, which we view as a form of deep information fusion occurring within the shared layers rather than only at the level of task heads.
>
> We agree that more explicit fusion mechanisms are an interesting direction, for example, designing regularizers or constraints that more directly inject ligand-based knowledge, such as selectivity profiles, into the structural branch. However, such approaches would typically require an additional mapping from high-level properties (e.g., selectivity) onto specific geometric or residue-level features on the protein, which is nontrivial. By contrast, our current design leverages both structure-based and ligand-based data “as is,” allowing the model to learn this alignment implicitly through its shared representation space.
>
> 3. Discussion of ablation results:
>
> We have moved the ablation study into the main paper and extended its discussion.
>
> a. Punctuation:
>
> Thank you for pointing this out. We have gone over all equations and fixed the punctuation.
>
> b. Discussion of binding site prediction results:
>
> This omission was primarily due to page limits and because we treated the binding site prediction task mainly as a sanity check. However, we recognize that this does not justify the lack of analysis, and we have now added a brief discussion in Section 4.3, Binding Site Prediction.
>
> c. Bolding criterion in virtual screening tables:
>
> As indicated in the captions of Tables 1, G1, and G2: “The best value per column is marked in bold; values within the mean absolute deviation of the best are also highlighted.” We are the only ones to report re-runs and provide error bars for the virtual screening experiments. When our method achieves the best value but another model’s result falls within our error bars, we consider the two methods to be equivalent; similarly, if the best model’s value lies within our error bars, we also consider our method to be on par. Thus, two values may appear bolded in the same column.
>
> We hope that our responses and the additional clarifications address all of the reviewer’s questions and concerns.

---

### Official Review · Reviewer_fBpy · 2025-11-01

**Soundness:** 2
**Presentation:** 3
**Contribution:** 2
**Rating:** 2
**Confidence:** 5

**Summary:**

### Summary
The paper presents ConGLUDE, a unified framework that combines structure-based and ligand-based drug discovery via geometric contrastive learning. It co-trains a protein encoder (for both binding-pocket and global representations) and a ligand encoder on large structural (PDBbind) and bioactivity datasets. The model achieves state-of-the-art zero-shot virtual-screening results on DUD-E and LIT-PCBA, outperforming other methods on some metrics.

**Strengths:**

---

### Strengths
- Clear unification: Bridges SBDD and LBDD with a simple, well-motivated contrastive framework.
- Scalable and generalizable: Handles diverse drug-discovery tasks (screening, pocket prediction, target fishing).
- Timely contribution: Moves toward foundation-model paradigms for computational drug design.

**Weaknesses:**

---

### Weaknesses
- Limited empirical advantage: The reported results do not consistently surpass other ML-based benchmarks. For example, in virtual screening, DrugCLIP achieves comparable results on DUD-E, despite being trained on less data and without ligand-based datasets.
- Low methodological novelty: The structure encoder is adopted from prior work, while the ligand encoder consists of only a few MLP layers. The contrastive loss also appears to be heavily inspired by DrugCLIP.
- Missing baselines: The ligand-conditioned pocket selection task should be compared with stronger baselines, such as those in the DiffDock paper or all-atom structure prediction models like AlphaFold3. Additionally, no baseline method is provided for the target-fishing task.
- Unclear characterization of ligand-based design: Ligand-based drug design typically applies to cases where target structures are unavailable or unknown, relying on 2D/3D similarity of known active molecules. However, in this paper, the authors still use protein representations during “ligand-based training,” which may not fully align with that definition.

**Questions:**

### Questions
- Why not encode small molecules as graphs? This is a more common choice in molecular representation learning.
- How does the speed of this method compare to DrugCLIP?
- Since the data split is based on protein sequences, how similar are the most similar molecules between the training and test sets?

---

> ### Author Response · Authors · 2025-11-28
> **Response to Reviewer fBpy (part 1)**
>
> We thank the reviewer for recognizing the motivation, generalizability, and timeliness of our contribution. We also appreciate the suggested experiments, which will strengthen our paper, and we address all raised weaknesses and questions in detail below.
>
> **Weaknesses**
>
> 1\.     Limited empirical advantage:
>
> While the gains on individual benchmarks may appear modest, as we highlight in the general response, ConGLUDe is the only method that achieves strong and consistent performance across *both* DUD-E and LIT-PCBA. Specifically, we outperform DrugCLIP on LIT-PCBA and SPRINT on DUD-E, demonstrating that our unified approach generalizes more robustly across heterogeneous tasks than methods specialized for a single data modality. It is worth noting that DUD-E inherently seems to benefit from training on structure-based data only, which is supported by our ablation study (Table 5), which shows that training ConGLUDe on structure-based data only yields strong performance on DUD-E but at the cost of losing generalizability to more realistic drug discovery datasets such as LIT-PCBA.
>
> Moreover, ConGLUDe stands out from other methods by being capable of handling multiple tasks. This is further demonstrated by the target fishing results, which show a clear empirical advantage compared to all baseline methods (see Table 2 and Weakness 3).
>
> 2\.     Methodological novelty:
>
> We agree that our model builds on established components, but as detailed in our general response, the way these components are integrated is fundamentally different from prior work. Critically, this architectural setup is what *enables combined training* on both structure-based and ligand-based data while still modeling pocket-level interactions. In contrast, existing approaches either rely on explicit pocket information and thus operate only on structure-based data or, when trained on ligand-based data, are unable to provide pocket-specific insights. For completeness and context, we refer the reviewer to the general response, where these points are discussed in more detail.
>
> 3\.     Missing baselines:
>
> You are correct that some important baselines were missing. We have now added several additional baselines, particularly for pocket selection and target fishing. The corresponding results can be found in Tables 2 and 4\.
>
> a)     We included **P2Rank+DrugCLIP** and **VN-EGNN+DrugCLIP** for all tasks as a two-step pipeline of pocket prediction followed by encoding using DrugCLIP. For the pocket selection task, we use P2Rank and VN-EGNN to propose candidate pockets and then rank them using the similarity between the DrugCLIP-encoded pocket and ligand, which does not improve over the pocket-prediction baselines alone. We also apply this pipeline to target fishing, where the true binding pockets are unknown, and DrugCLIP’s performance here is close to random. Finally, we add this baseline to the virtual screening results to show that DrugCLIP heavily benefits from prior knowledge of the binding pocket, which is not available in many realistic scenarios.
>
> b)     We added **SPRINT** as a baseline for target fishing \- its performance in this setting is poor. It cannot be used for pocket selection because it does not incorporate pocket information.
>
> c)     We evaluated **DiffDock** for target fishing \- while it is the only baseline that achieves performance significantly above random, it is still clearly outperformed by ConGLUDe. In principle it could be applied to virtual screening as well, but this would be prohibitively expensive, as the target fishing experiment alone required over 12 GPU days.
>
> d)     **Boltz-2** is included as a representative all-atom co-folding method because, unlike AlphaFold3, it is fully open-source while still offering state-of-the-art co-folding performance. For pocket selection on the PDBBind time-split test set, Boltz-2 achieves a DCC success rate of 0.69, clearly outperforming all other methods (including the previous best value of 0.47 achieved by ConGLUDe). However, its training data includes all PDB structures up to 06/01/2023 \[1\], which covers the entire test set, explaining the strong performance. On ASD, evaluation is still in progress, but we expect similarly strong results given the substantial train-test overlap. Each prediction takes about 20 seconds after pre-processing, making large-scale virtual screening and target fishing over more than a million protein-ligand pairs computationally infeasible.

---

> > ### Author Response · Authors · 2025-11-28
> > **Response to Reviewer fBpy (part 2)**
> >
> > 4\.     Unclear characterization of ligand-based design:
> >
> > We agree that our use of the term LBDD was somewhat imprecise. You are correct that, in the strict sense, LBDD refers to settings without protein structures. However, our approach does integrate data traditionally associated with both SBDD and LBDD, and for ligand-based datasets without annotated PDB structures, AlphaFold predictions can provide the necessary protein representations. To avoid confusion, we have replaced the term “ligand-based training” with “training on ligand-based data” throughout the paper, which more accurately reflects our setting, and we have updated the last paragraph of the introduction to clarify this distinction.
> >
> > **Questions**
> >
> > 1. Small molecules as graphs:
> >
> > We opted for molecular fingerprints and a simple MLP instead of graph representations and graph neural networks, as they are faster and many QSAR studies have shown fingerprint-based encoders to achieve better performance \[2, 3\].
> >
> > 2\.     Speed comparison with DrugCLIP:
> >
> > We evaluated the inference time of DrugCLIP and ConGLUDe in the virtual screening setting on the DUD-E dataset. The average per-protein timings (with standard deviations) after preprocessing are:
> >
> > DrugCLIP: 47.2s ($\\pm$ 35.4)
> > ConGLUDe: 0.9s ($\\pm$ 0.7)
> >
> > The substantial speed-up of ConGLUDe is likely due to its lightweight molecule encoder, which allows simultaneous encoding of a large number of ligands, in contrast to the 3D-transformer-based architecture used in DrugCLIP. It is also worth noting that, compared to docking or co-folding approaches, both DrugCLIP and ConGLUDe are faster by several orders of magnitude, enabling large-scale virtual screening.
> >
> > 3\.     Molecule similarity:
> >
> > We added Figure D1 in the appendix, which shows the distribution of maximum ECFP Tanimoto similarities between each test molecules and all training molecules. Although some molecules overlap or are very similar, especially between the ligand-based training data and the LIT-PCBA test set, for most test molecules, even the closest training molecule has a similarity below 0.7. We used the protein sequence-based splits for two reasons: (a) to remain consistent with the splits used by our baseline methods, and (b) because the primary application, virtual screening on new targets using existing compound libraries, typically involves such molecule overlap at inference time, making this level of overlap acceptable for evaluation purposes.
> >
> > We believe that our clarifications, the inclusion of additional baselines, and the updated analyses address the concerns raised, and we hope the reviewer will consider revising their overall evaluation in light of these improvements.
> >
> > \[1\] Passaro et al. Boltz-2: Towards Accurate and Efficient Binding Affinity Prediction. *bioRxiv* (2025)
> >
> > \[2\] Jiang et al. Could graph neural networks learn better molecular representations for drug discovery? A comparison study of descriptor-based and graph-based models. *J Cheminform* (2021)
> >
> > \[3\] Ebner et al. Measuring AI Progress in Drug Discovery: A Reproducible Leaderboard for the Tox21 Challenge. *arXiv* (2025)

---

### Official Review · Reviewer_Gz1z · 2025-11-01

**Soundness:** 3
**Presentation:** 3
**Contribution:** 2
**Rating:** 4
**Confidence:** 5

**Summary:**

The paper presents ConGLUDe (Contrastive Geometric Learning for Unified Computational Drug Design), a framework that unifies structure-based (SBDD) and ligand-based (LBDD) drug design. It jointly trains a geometric protein encoder and a ligand encoder using contrastive learning on both 3D protein–ligand complexes (PDBbind) and large-scale bioactivity datasets (ChEMBL, PubChem, BindingDB). ConGLUDe performs multiple drug discovery tasks—including virtual screening, binding pocket prediction, ligand-conditioned pocket selection, and target fishing—within one model. Experiments show state-of-the-art or competitive results across benchmarks, demonstrating effective integration of structure and ligand data. The method’s limitations include reliance on available 3D protein structures and chemical space coverage, but it represents a step toward foundation models for drug discovery

**Strengths:**

The paper is clearly written, and the description of the proposed model is well articulated. I agree with the idea of leveraging both large-scale protein data and extensive molecular data for modeling interactions.

**Weaknesses:**

1. Although the authors emphasize the goal of unifying large-scale protein and molecular data to train a unified model, I believe their proposed method does not fully achieve this. In practice, the model is mainly trained on PDBBind or BindingDB or CHEMBL, containing protein–ligand pairwise data. Such datasets are far smaller in scale than purely protein datasets (e.g., UniProt) or purely molecular datasets. Therefore, in terms of data utilization, the approach is not fundamentally different from previous methods.
2.	Regarding the model, the main innovation lies in improving the protein encoder to simultaneously encode both the pocket and the protein. However, this is essentially an enhancement of the encoder design rather than a change in the overall training paradigm. Architecturally, the structure-based training remains similar to other contrastive learning–based approaches, the ligand-based training only adds a new loss term and dataset, without introducing a novel architecturally.
3.	The virtual screening results appear relatively marginal, performing roughly on par with DrugCLIP and SPRINT on two benchmark datasets.

**Questions:**

1.	The authors should show that their adaptive pocket–encoder approach outperforms the two-step method (e.g., Fpocket + pocket encoder) to justify the value of unifying protein and pocket representations in one model.
2.	How do the authors view GNN/EGNN models’ ability to encode geometry, particularly the trade-off between equivariance and efficiency? Could data augmentation replace strict equivariance to enable usage of more powerful architectures?

---

> ### Author Response · Authors · 2025-11-28
> **Response to Reviewer Gz1z**
>
> Thank you for the constructive feedback, including the suggested additional baseline, and for highlighting both the clarity of our paper and the value of leveraging large-scale protein and molecular data for interaction modeling. In the following, we address each of the points raised as “Weaknesses” and “Questions” in detail.
>
> **Weaknesses**
>
> 1\.      Unifying protein and molecular data:
>
> We appreciate the reviewer’s point and agree that we do not incorporate purely protein-only or molecule-only datasets, as learning protein-ligand interactions inherently requires *paired* data. Our contribution is to unify the two major classes of paired data used in drug discovery: 3D structures of protein-ligand complexes (structure-based, e.g., PDBBind) and large-scale ligand-based bioactivity data (e.g., ChEMBL, PubChem, BindingDB). Our model is trained *simultaneously* on both types of data.
>
> 2\.     Novelty:
>
> We agree that our model builds on established components, but as detailed in our general response, the way these components are integrated is fundamentally different from prior work. Critically, this architectural setup is what *enables combined training* on both structure-based and ligand-based data while still modeling pocket-level interactions. In contrast, existing approaches either rely on explicit pocket information and thus operate only on structure-based data or, when trained on ligand-based data, are unable to provide pocket-specific insights. For completeness and context, we refer the reviewer to the general response, where these points are discussed in more detail.
>
> 3\.     Virtual screening performance:
>
> While the gains on individual benchmarks may appear modest, as we highlight in the general response, ConGLUDe is the only method that achieves strong and consistent performance across *both* DUD-E and LIT-PCBA. Specifically, we clearly outperform DrugCLIP on LIT-PCBA and SPRINT on DUD-E, indicating that our unified approach *generalizes more robustly* across heterogeneous tasks than methods specialized for only one data modality. In addition, the newly included baseline experiments for target fishing (see Table 2\) show that ConGLUDe successfully generalizes to this setting, whereas nearly all compared approaches (including DrugCLIP and SPRINT) perform close to random.
>
> **Questions**
>
> 1\.    Comparison to two-step method (pocket prediction \+ pocket encoder):
>
> This is a very good suggestion. We have incorporated two such baselines across all tasks (virtual screening, target fishing, and pocket selection). Since the DrugCLIP pocket encoder is conceptually closest to our approach, we pair it with two pocket predictors: P2Rank, which is widely used for pocket prediction, and VN-EGNN, based on the predictor underlying ConGLUDe.
>
> Introducing a separate pocket prediction step substantially reduces DrugCLIP’s performance on virtual screening (see Appendix Tables G1 and G2). For target fishing, where pocket prediction is required because the binding pockets of test proteins are unknown, DrugCLIP performs close to random with both predictors (see Table 2). Finally, in the pocket selection task, ranking pockets by ligand similarity using DrugCLIP encoders does not improve over the pocket prediction baselines alone (see Table 4).
>
> 2\.     Equivariance vs efficiency:
>
> The reviewer raises an interesting point about the capacity of GNN/EGNN models, thanks! Indeed, we found that EGNN is able to capture the geometry of proteins well, which is shown by the fact that it is very accurate and precise at detecting binding pockets. Higher-order models, such as MACE [1] are computationally not feasible, such that EGNN appears as the best choice within this category of methods. A more general MPNN could indeed be used, and the equivariances could be learned through data augmentation (by rotating and shifting the protein). Indeed, both directions, a) baking equivariances into the architectures and b) learning equivariances, are currently successful paradigms in machine learning. In the regime of this work, in which both data availability and computation power is on the lower side, the equivariance approach is more promising, which is why we settled on VN-EGNN.
>
> We hope that our responses and the inclusion of the suggested additional baseline fully address all raised questions and will encourage the reviewer to consider raising their overall evaluation.
>
> [1] Batatia et al. MACE: Higher Order Equivariant Message Passing Neural Networks for Fast and Accurate Force Fields. *NeurIPS* (2022)

---

### Author Response · Authors · 2025-11-14
**General Response to the Reviewers**

First of all, we want to thank the reviewers for their constructive feedback and are truly grateful for the thoughtful suggestions, which will help us further strengthen our paper. In particular, we appreciate that the reviewers recognized the power of integrating SBDD and LBDD within a single framework and valued the broad applicability of our model across many different drug discovery tasks.

We will provide a full set of point-by-point responses and additional experiments in the upcoming full rebuttal. For now, however, we already want to address two recurring concerns raised by multiple reviewers that appear to result from insufficient clarity in our presentation rather than fundamental limitations of our work:

**1.) Technical novelty:**

While our model does build on established components (an adapted VN-EGNN as the protein encoder, a simple MLP-based ligand encoder, and a CLIP-style contrastive loss), we use these components in ways that are distinct from prior work:
* We are, to the best of our knowledge, the first to use VN-EGNN as a general-purpose protein and binding site feature encoder, rather than solely as a pocket prediction model whose final representations are not used for downstream tasks.
* We uniquely apply the CLIP-style loss directly to a set of pocket representations on a single protein and for a given ligand, enabling the model to select the most likely binding pocket - a capability not explored in prior work.

More importantly, the **integration of these components is what allows the *combined* training on structure- and ligand-based data** in the first place, unlocking capabilities not achievable by prior methods: ConGLUDe can simultaneously predict whether a ligand binds to a protein and infer the most likely binding pocket, without requiring prior pocket information. In contrast, existing methods either require the pocket as input (e.g., DrugCLIP) and are thus limited to structure-based data, or provide no binding site information at all (e.g., SPRINT). Our framework bridges this gap and offers functionality neither approach can achieve alone.

**2.) Performance and generalization:**

Although improvements on individual benchmarks may appear modest when viewed in isolation, our model demonstrates consistently strong performance across diverse datasets and tasks. Notably, it is the *only* method to achieve state-of-the-art performance on *both* DUD-E, a dataset focused on structural complementarity, and LIT-PCBA, which captures broad ligand-activity patterns from HTS assays. This illustrates that our unified method effectively integrates structural and ligand-based information, enabling **robust generalization across heterogeneous tasks**.

Furthermore, we show strong generalization capabilities of ConGLUDe to a new target fishing dataset with a distinct biological curation methodology. Baseline comparisons for these results will be provided in the full rebuttal.

---

### Author Response · Authors · 2025-11-28
**General Response 2 - Full Rebuttal**

We have now uploaded the full rebuttal, including detailed responses to all questions and concerns raised by the reviewers. We thank the reviewers for their patience, as several of the requested additional experiments required considerable time for both setup and computation.

To facilitate verification of all modifications, we provide (i) a revised manuscript with all edits highlighted in the supplementary material and (ii) an updated clean version as the main submission.

In response to the reviewers’ requests, we substantially expanded our experimental evaluation:

* We added baseline results for DrugCLIP, SPRINT, and DiffDock in the target fishing experiment (Table 2\) and show that ConGLUDe outperforms all of them by a substantial margin.

* We included new baselines that pair DrugCLIP with a pocket prediction method (P2Rank or VN-EGNN), enabling a two-step pipeline that mirrors the ConGLUDe process for pocket selection and virtual screening (Table 4 and Tables G1–G2).

All new results and discussions are included in both the revised manuscript and the point-by-point responses.

We kindly invite the reviewers to examine our detailed responses and consider the additional analyses and revisions in their evaluations.

---

### Author Response · Authors · 2025-11-30
**Rebuttal Summary for AC**

Dear new AC,

We anticipate you will have a lot to read in the coming days, so we provide a concise version of our rebuttal here.

Across the reviews, we identified three primary concerns, which we address in the responses to the reviewers and in the revised manuscript:

* Technical novelty. While our model uses established components, their integration is fundamentally different from prior work. Our architecture is the first to enable joint training on both structure-based and ligand-based data while still modeling pocket-level interactions, whereas existing methods either require explicit pocket information, i.e. can only handle structure-based data, or cannot provide pocket-specific insights at all.

* Performance and generalization. Although per-benchmark gains may appear modest, ConGLUDe is the only method that performs strongly and consistently across both virtual screening benchmarks, the pocket selection benchmarks, and outperforms other approaches on the target fishing task (see next point).

* Missing baselines. We added baseline results for DrugCLIP, SPRINT, and DiffDock in the target fishing experiment and found that ConGLUDe outperforms all of them by a substantial margin. We further added two-step baselines that couple DrugCLIP with pocket predictors (P2Rank or VN-EGNN) for pocket selection and virtual screening, where our unified model also performs better.

---

### Meta-Review · Area_Chair_gWRS · 2026-01-04

**Summary:**

The paper proposes ConGLUDe, a framework designed to unify SBDD and LBDD. The method employs a Contrastive Geometric Learning approach, utilizing a VN-EGNN encoder for proteins/pockets and an MLP for ligands. The model is trained via contrastive loss on a combination of 3D complex data and bioactivity data. The authors evaluate the model on virtual screening, binding site prediction, and target fishing.

While the motivation to unify these two paradigms is strong, the consensus among reviewers is that the technical execution relies on a relatively standard assembly of existing components rather than a novel fusion mechanism. Furthermore, the empirical results are mixed; while the method performs well on LIT-PCBA and the newly added target fishing baseline, it offers marginal improvement over existing baselines like DrugCLIP on DUD-E, and ablation studies reveal that the co-training can sometimes degrade performance compared to structure-only training.

**Reviewer Concerns:**

## Addressed:
- The authors added requested baselines for target fishing (DrugCLIP, SPRINT, DiffDock) and two-step pocket selection methods (P2Rank/VN-EGNN + DrugCLIP). These results  demonstrate ConGLUDe's advantage in the specific task of zero-shot target fishing.

- The authors corrected the terminology to "training on ligand-based data" and clarified that they use AF structures when experimental ones are missing.

- The authors provided inference time comparisons showing ConGLUDe is significantly faster than DrugCLIP.

- The mismatch in Figure 2 wrt loss terms was corrected.

## Outstanding:

- The architecture is fundamentally a combination of VN-EGNN and a standard contrastive loss. The unification is achieved through alternating training batches rather than a novel architectural fusion. Reviewer Gz1z noted this is "not fundamentally different from previous methods," and the rebuttal's argument that the application of these components is novel is not convincing enough.

- On the DUD-E benchmark, ConGLUDe does not convincingly outperform DrugCLIP. Reviewer fBpy noted that DrugCLIP achieves comparable results without needing the massive ligand-based dataset. The rebuttal confirms that ConGLUDe beats DrugCLIP on LIT-PCBA but only matches it on DUD-E.

-  Reviewer FPHC pointed out that the structure-only model outperforms the unified model on DUD-E. The authors argue this is because DUD-E favors structural complementarity, but this admission undermines the central premise of the paper and becomes a question that this unification may not be optimal.

**Reviewer Scores:**

Reviewer Gz1z 4==> 4 (unchange). While the baseline questions were answered, the fundamental concern about the method being an incremental enhancement of VN-EGNN via multi-task training remains unaddressed.

Reviewer fBpy 2 ==> 2 or 4 (unchange or slightly increase). The reviewer would likely acknowledge the added baselines and speed comparisons, but the fact that the method fails to beat DrugCLIP on DUD-E despite using significantly more data would prevent a move to acceptance.

Reviewer FPHC 6 ==> 4 (slightly decrease). This reviewer was the most positive but expressed concern that the result somewhat undermines the core argument that "fusing both data types is crucial." In the main text, the author admit that "sructure-based data are critical for performance, while ligand-based data are not strictly necessary and, in fact, removing them even slightly improves results". This reveal that the fusion is not synergistic for all tasks

Reviewer ZA3q 4 ==> 4 (unchange). The disjointed feeling of the paper and the limited novely and marginal performance gain remain valid concerns.

---

### Decision · Program_Chairs · 2026-01-26

Reject